# Fast Channel Simulation via Error-Correcting Codes

**Sharang M. Sriramu**
School of ECE
Cornell University
Ithaca, NY 14853
sms579@cornell.edu

**Rochelle Barsz**
School of ECE
Cornell University
Ithaca, NY 14853
rsb359@cornell.edu

**Elizabeth Polito**
School of ECE
Cornell University
Ithaca, NY 14853
emp234@cornell.edu

**Aaron B. Wagner**
School of ECE
Cornell University
Ithaca, NY 14853
wagner@cornell.edu

## Abstract

We consider the design of practically-implementable schemes for the task of channel simulation. Existing methods do not scale with the number of simultaneous uses of the channel and are therefore unable to harness the amortization gains associated with simulating many uses of the channel at once. We show how techniques from the theory of error-correcting codes can be applied to achieve scalability and hence improved performance. As an exemplar, we focus on how polar codes can be used to efficiently simulate i.i.d. copies of a class of binary-output channels.

## 1 Introduction

*Channel simulation* refers to the task in which Alice observes a realization $x$ of a random variable $X$ and sends a bit string to Bob. Bob, who shares common randomness with Alice, outputs a random variable $Y$ using both the message and the common randomness. The goal is to minimize the length of the bit string subject to the constraint that $Y$ should have a specified distribution $P_{Y|X}(\cdot|x)$. This problem can be viewed as a "soft" or "stochastic" generalization of quantization. As with quantization, $Y$ has a digital representation (through Alice's message), and it can be viewed as a degraded version of $X$. The difference is that here the degrading process is stochastic in general. In fact, the quantization problem is subsumed by taking the channel $P_{Y|X}$ to be deterministic.

This stochastic generalization of quantization arises in lossy compression of various types of sources including images [Flamich et al., 2020, Ballé et al., 2020], models [Havasi et al., 2018], and gradients [Shah et al., 2022]. In these applications, $X$ often represents a vector of latent variables, model weights, or even an image consisting of millions of pixels [Theis et al., 2022]. The channel of interest is therefore high dimensional, and it is usually independent across the dimensions. For conventional quantization, it has long been recognized that the optimum rate-distortion tradeoff is more favorable in higher dimensions [Cover and Thomas, 2006], a trend that one expects to generalize to channel simulation. Indeed, let $n \cdot R_n$ be the minimum number of bits required to generate $Y^n = (Y_1, \ldots, Y_n)$, when $X^n$ is i.i.d. and $Y^n$ is conditionally i.i.d. given $X^n$. Thus $R_n$ is the minimum number of bits per dimension when simulating the channel $n$ times. The sequence $nR_n$

38th Conference on Neural Information Processing Systems (NeurIPS 2024).

can be shown to be *subadditive* and therefore satisfies (e.g., Liggett [1999, Thm. B22])

$$\lim_{n \to \infty} R_n = \inf_n R_n. \tag{1}$$

It is known that $R_1$ satisfies [Li and El Gamal, 2018]

$$I(X;Y) \leq R_1 \leq I(X;Y) + \log(I(X;Y) + 1) + 5, \tag{2}$$

where $I(X;Y)$ refers to conventional Shannon mutual information. Applying this to i.i.d. $(X^n, Y^n)$ and using the fact that $I(X^n; Y^n) = nI(X;Y)$, we have

$$nI(X;Y) \leq n \cdot R_n \leq nI(X;Y) + \log(nI(X;Y) + 1) + 5, \tag{3}$$

which shows that as $n \to \infty$, $R_n$ approaches the lower bound $I(X;Y)$. The challenge, for both quantization and channel simulation, is that the complexity of schemes tends to grow exponentially in $n$. In fact, although many channel simulation schemes have been proposed [Harsha et al., 2007], [Li and El Gamal, 2018], [Flamich et al., 2022], [Flamich and Theis, 2023], [Flamich et al., 2024], none have complexity that scales subexponentially in $n$, ignoring isolated examples for which $R_1$ happens to equal $I(X;Y)$ [Zamir and Feder, 1992], [Agustsson and Theis, 2020].

Vector quantization has long been recognized to be the dual, in a precise sense, of channel coding [Pradhan et al., 2003]. In channel coding, the decoder maps an arbitrary point to an element of a finite set that is "close" in some channel-dependent sense. This is analogous to the role of the encoder in quantization. Likewise, the encoder in channel coding is analogous to the decoder in quantization: both map bit strings to elements of said discrete set. Thus new techniques for channel coding can often be applied to vector quantization [Goblick, 1963], [Viterbi and Omura, 1974] and vice versa [Laroia et al., 1994].

The goal of this paper is to demonstrate how ideas from coding theory can be applied to the channel simulation problem. We shall see that by adopting these techniques, we can develop schemes that significantly outperform state-of-the art methods, both in terms of their scalability and their rate performance. Specifically, we show how *polar codes* [Arikan, 2009] can be applied to the simulation problem using a method called `PolarSim`. Polar codes make for a good exemplar of this general proposal for five reasons. First, they have excellent channel coding performance, both theoretically [Mondelli et al., 2016] and practically [Egilmez et al., 2019]. Second, their complexity scales as $n \log n$. Third, they require no manual tuning. Fourth, they are simple to describe, requiring minimal background in coding theory. Finally, there exist highly optimized implementations of the encoding and decoding algorithms (e.g., [Pfister, 2023]). Their limitation is that, in their basic form, they can only be applied to symmetric binary-input channels (see (4) to follow). As we shall see, this means that `PolarSim` can only simulate symmetric binary-output channels. This class includes, for example, the binary symmetric channel, the (reverse) binary erasure channel, and channels of the form $X \to \mathrm{signum}(X + Z)$, where $X$ and $Z$ are real-valued, independent random variables with symmetric distributions. Note that the input to the channel need not be binary and may even be continuous.

For these channels, we show both theoretically and experimentally that, by scaling up the dimension, the rate of `PolarSim` can be made to approach the mutual information lower bound $I(X;Y) \leq R_n$ from (3). The superior scalability of `PolarSim` thus translates to a significant rate improvement over the state-of-the-art, since those schemes are not able to harness the amortization gain associated with letting $n$ grow. Although `PolarSim` is restricted to binary-output channels, it is worth noting that there are currently no known schemes that simulate any nontrivial class of channels with even subexponential complexity in $n$. Also, for compression applications, a binary output alphabet is not unreasonable. It should be emphasized that the binary-output restriction is particular to the basic form of polar codes, not error-correction methods in general. In the supplementary materials we discuss how a different coding technique, trellis coded modulation, can be applied to closely simulate a Gaussian channel. See also the discussion in the Concluding Remarks section on non-binary polar codes.

One lesson from the coding theory literature, especially with the advent of modern coding theory in the 1990s [Richardson and Urbanke, 2008], is that it is advantageous to prioritize scalability with the dimension $n$ over achieving optimal performance for particular $n$. The reason is that performance naturally improves with increasing $n$ (as in (1)-(3) above), and this improvement can overcome suboptimality at any given value of $n$. For state-of-the-art channel codes, the decoder typically does not implement the optimal (maximum likelihood) decision rule. Instead, it implements a

scalable approximation to it. This design strategy of favoring scalability over optimality is now well established in coding theory and vector quantization, and our goal here is to show how it can be profitably applied to channel simulation.

The rest of the paper is organized as follows. We provide the necessary background on polar codes in Section 2. Section 3 describes the `PolarSim` scheme, with Section 3.3 containing the theoretical result and Section 3.4 contains simulation results. Some concluding remarks are offered in Section 5.

## 1.1 Terminology and Notation

We follow the standard convention of denoting the dimensionality of vectors by their superscript. We will also denote compound i.i.d. channels by superscripts: $p^{\times n}$ denotes $n$ i.i.d. copies of a distribution $p$. The number of copies here is referred to as the *block length* or *dimension* of the channel. This is to be distinguished from the notation $F^{\otimes n}$ which denotes a $n$-fold self-Kronecker product of a tensor $F$. All logarithms mentioned in this paper are base 2. The *binary entropy function* $h_B : [0, \frac{1}{2}] \mapsto [0, 1]$ is defined as $h_B(p) = -p \log p - (1 - p) \log(1 - p)$. We will also refer to its inverse $h_B^{-1} : [0, 1] \mapsto [0, \frac{1}{2}]$ defined such that $h_B^{-1}(h_B(p)) = p$.

# 2 Background

## 2.1 The Channel Simulation Problem

Consider a joint probability measure $p_{XY}$ on the set $\mathcal{X} \times \mathcal{Y}$. Alice receives a sequence of $n$ symbols from the input alphabet $\mathcal{X}$ drawn according to $p_X^{\times n}$ and encodes it into a binary string that she transmits to Bob. Upon receiving the message from Alice, Bob then decodes it to generate a sample from the channel $p_{Y|X}^{\times n}$. The objective is to find coding schemes that minimize the average *rate*— i.e., the average amortized length of the bit string transmitted by Alice. We refer to $n$ as the *block length* or the *dimension*. We require that the set of strings that Alice can transmit to Bob to form a *prefix-free* set, meaning that no string in the set is a prefix of any other. Alice's message is thus self-terminating, and schemes for block length $m$ and $n$ can be combined to obtain a scheme for block length $m + n$ by concatenation. If $R_n$ denotes the minimum average rate, i.e., the minimum average length of Alice's string over all schemes, normalized by $n$, then $nR_n$ is subadditive in $n$, as noted earlier.

Both Alice and Bob are permitted to use randomized strategies and are assumed to share a source of common randomness. Under this assumption, Li and El Gamal [2018] prove the performance bounds (2) and (3) above (see also Harsha et al. [2007]). For large $n$, Sriramu and Wagner [2024] improve upon this result for i.i.d. discrete memoryless channels, showing that the logarithmic redundancy term can be halved for some channels and eliminated for all others. While these schemes are nearly rate-optimal, their complexity scales exponentially in $n$. Other practical schemes have been proposed, although none have even subexponential scaling in $n$ outside the small class of channels for which the lower bound in (3) is tight for all $n$.

## 2.2 Background on Polar Codes

Modern coding theory has focused on the search for codes whose rates approach the theoretical limit while having a low encoding and decoding complexity. *Polar codes* are among the crowning achievements of this search. In their basic form, they are capacity-achieving linear codes for binary input channels $W_{X|Y}(\cdot|\cdot)$ satisfying the following symmetry condition: there exists a bijection $\pi : \mathcal{X} \mapsto \mathcal{X}$ such that $\pi^{-1} = \pi$ and

$$W(x|0) = W(\pi(x)|1) \text{ for all } x \in \mathcal{X}. \tag{4}$$

As a linear code, the encoding procedure for polar codes is defined by a generator matrix $G_n$ that maps an input $U^n \in \{0, 1\}^n$ to the channel input $Y^n$ as $Y^n = U^n G_n$ with all operations performed over $\mathbb{F}_2$. The generator matrix of size $n = 2^m$ can be constructed recursively from a kernel matrix $F = \left[\begin{smallmatrix} 1 & 0 \\ 1 & 1 \end{smallmatrix}\right]$ and the "bit-reversal" permutation matrix $B_n$—see Arikan [2009]. The generator matrix is then defined as $G_n = B_n F^{\otimes n}$. The structure of the generator matrix allows for encoding in $n \log n$ time.

Decoding proceeds by sequentially guessing each bit of $U^n$ in order. For this, one considers the *subchannel* $U_i \to (U^{i-1}, X^n)$. Specifically, given the realization of the output $x^n$ and the decoding

decisions for the prior bits $\hat{u}^{i-1}$, one computes the likelihood ratio

$$\frac{\Pr(U_i = 1|X^n = x^n, U^{i-1} = \hat{u}^{i-1})}{\Pr(U_i = 0|X^n = x^n, U^{i-1} = \hat{u}^{i-1})}, \tag{5}$$

and selects $\hat{U}_i$ accordingly. This decoding rule is suboptimal but scalable in that the likelihood ratio can be computed in $\log n$ steps using the recursive structure of $G_n$ [Arikan, 2009], resulting in decoding complexity of $n \log n$.

The suboptimality turns out to be acceptable because as $n$ grows, the subchannels *polarize*, meaning that for most $i$ the likelihood ratio in (5) is close to 0, 1 or $\infty$ with high probability. Equivalently, the mutual information $I(U_i; X^n, U^{i-1})$ is close to zero or one. Figure 1 illustrates this phenomenon for a *binary symmetric channel* (BSC) which is defined by $X = Y \oplus Z$, where $Z \sim \text{Bernoulli}(p)$ for some crossover probability $p \in [0, 1]$.

For communication, the encoder uses the "clean" subchannels with mutual information close to 1 to send information bits. The inputs to the remaining noisy channels, called the *frozen bits*, are fixed ahead of time and known to both the encoder and decoder. The data rate is thus the fraction of subchannels that are clean, which can be shown to approach capacity [Arikan, 2009].

Implementing the code requires determining which indices correspond to the clean subchannels. This can be done using the dimensionality reduction technique of Tal and Vardy [2013], by exploiting the recursive structure of the polar transform [Zhang et al., 2014, Arikan, 2009], or by Monte Carlo simulation.

## 3 Simulating Binary Output Channels

Previously, we described polar codes for constructing channel codes for binary *input* channels. Based on the duality between channel coding and channel simulation, we will see that this naturally leads to a scheme for simulating binary *output* channels.

We begin by describing a "toy scheme". This is not a rate-efficient scheme in its own right, but it serves as a foundation for `PolarSim`.

### 3.1 Toy Scheme for Binary Output Channel Simulation

Consider a joint distribution $p_{XY}$ where $p_Y$ is $\text{Bernoulli}\left(\frac{1}{2}\right)$. Then, the following algorithm simulates $p_{Y|X}$ exactly:

1. Use the common randomness to generate $Z \sim \text{Unif}(0, 1)$ and $V = \mathbf{1}\left(Z > \frac{1}{2}\right)$ at both the encoder and decoder.

2. At the encoder, having observed an input realization $x$, compute the output bit $Y = \mathbf{1}\left(Z > p_{Y|X}(0|x)\right)$ and the correction bit $\Delta = Y \oplus V$.

3. Transmit $\Delta$ to the decoder after lossless compression.

4. Recover $Y = \Delta \oplus V$ at the decoder.

In Appendix A, we show that the rate associated with repeated application of this scheme is upper bounded by $h_B\left(\frac{1}{2} - h_B^{-1}(1 - I(X; Y))\right)$. As we can see in Figure 2, this is highly suboptimal in general. However, we note that for the special case in which the mutual information is *polarized*, i.e., where $I(X; Y) \approx 0$ or $I(X; Y) \approx 1$, the toy scheme is close to optimal.

This observation is crucial as it suggests a path forward: If we can transform a given channel simulation problem to the problem of simulating polarized channels, it can be solved rate-efficiently using the toy scheme. Polar codes provide us with the means to achieve such a transformation.

### 3.2 Channel Simulation using Polar Codes

As in the toy scheme, we shall consider a joint distribution $p_{XY}$ where $p_Y$ is $\text{Bernoulli}\left(\frac{1}{2}\right)$. We will additionally assume that $p_{X|Y}$ satisfies the symmetry condition described in (4). Let us then examine

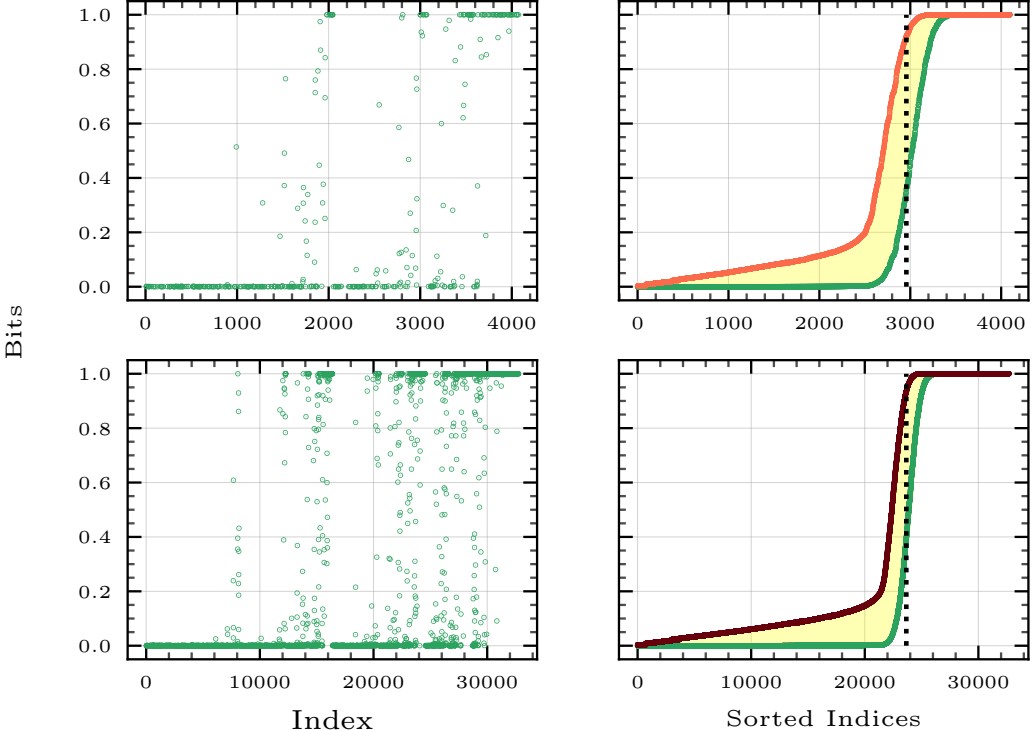

**Figure 1:** Channel polarization for a BSC with crossover probability 0.2 and block lengths $n = 2^{12}$ (**top**) and $n = 2^{15}$ (**bottom**). The scatter plots on the **left** show the subchannel capacities $I(U_i; X^n, U^{i-1})$ for each index $i$. In the curves (■) on the **right**, these indices are sorted in the increasing order of their subchannel capacities for better visualization. The area under these curves is the mutual information lower bounds at their respective block length. The **vertical dotted line** marks the ideal polarized channel, i.e., the fraction of indices to its right is equal to the mutual information of the channel. We see that the sorted subchannel capacity curve approaches this line as the block length is increased. Finally, we also plot the theoretical upper bound (see (38)) on the rate of our proposed scheme, PolarSim, for block lengths ■ $n = 2^{12}$ and ■ $n = 2^{15}$. The area under these curves is an upper bound on the rate of PolarSim. The shaded area in between is therefore an upper bound on the redundancy of PolarSim, which vanishes as $n \to \infty$ due to the polarization phenomenon.

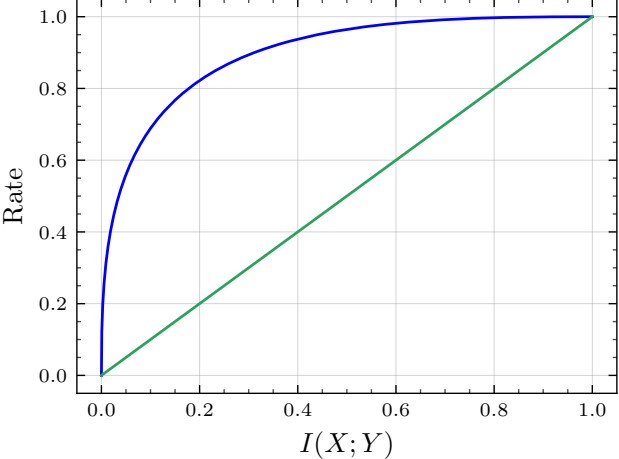

**Figure 2:** The upper bound on the rate of the toy scheme (■) described in section 3.1 is plotted against the mutual information lower bound (■).

the problem of simulating two independent realizations of the channel $p_{Y|X}$, $X^2 \to Y^2$. Consider the following bijection applied to the output $(Y_1, Y_2)$:

$$U_1 = Y_1 \oplus Y_2,$$
$$U_2 = Y_2. \tag{6}$$

It is clear that simulating the original pair of i.i.d. channels $X_1 \to Y_1$ and $X_2 \to Y_2$ is equivalent to simulating the transformed pair of channels $(X_1, X_2) \to U_1$ and $(X_1, X_2, U_1) \to U_2$.

The mutual information of each of the original i.i.d. channels are equal to $I(X; Y)$. However, the two transformed channels differ in terms of mutual information: $I(X_1, X_2, U_1; U_2) > I(X; Y)$ because $U_2$ is observed through two different channels. This necessitates that the other channel has lower mutual information: $I(X_1, X_2; U_1) < I(X; Y)$. Therefore, the linear transformation (6) we applied to the output had the effect of *polarizing* the target channel.

For block lengths that are larger powers of two, we can apply the transform inductively, resulting in the relationship we saw in section 2: $U^n = Y^n G_n^{-1}$. Similar to the two-dimensional ($n = 2$) case, this transforms the original simulation problem $X^n \to Y^n$ into the problem of simulating the *subchannels* $X^n \to U_1, (X^n, U_1) \to U_2, (X^n, U_1, U_2) \to U_3, \ldots, (X^n, U^{n-1}) \to U_n$. Arikan [2009] shows that these subchannels are polarized for large $n$: for each $i$, $I(U_i; X^n, U^{i-1}) \approx 0$ or $I(U_i; X^n, U^{i-1}) \approx 1$. This allows us to simulate them using the toy algorithm described in the previous section.

Algorithms 1 and 2 describe the complete scheme.

---

**Algorithm 1:** Encoder for simulating a channel using polar codes.

**Input** : Block length $n$
Random bit string $z^n \sim \text{Unif}\,([0, 1]^n)$
Probability table $\overline{p}^n \in [0, 1]^n$
Source string $x^n \in \mathcal{X}^n$
**Output** : String $b \in \{0, 1\}^*$
**for** $i = 1, \ldots, n$ **do**
  **if** $z_i > \texttt{SoftPolarDec}(x^n, u^{i-1})$ **then** $u_i \leftarrow 1$ **else** $u_i \leftarrow 0$
  **if** $z_i > 1/2$ **then** $v_i \leftarrow 1$ **else** $v_i \leftarrow 0$
  $\Delta_i \leftarrow u_i + v_i$
$b \leftarrow \texttt{Compress}(\Delta^n, \overline{p}^n)$
**return** $b$

---

**Algorithm 2:** Decoder for simulating a channel using polar codes.

**Input** : Block length $n$
Random bit string $z^n \sim \text{Unif}\,([0, 1]^n)$
Probability table $\overline{p}^n \in [0, 1]^n$
Compressed offset string $b$
**Output** : Simulated channel output $y^n \in \{0, 1\}^n$
$\Delta^n \leftarrow \texttt{Decompress}(b, \overline{p}^n)$
**for** $i = 1, \ldots, n$ **do**
  **if** $z_i > 1/2$ **then** $v_i \leftarrow 1$ **else** $v_i \leftarrow 0$
  $u_i \leftarrow \Delta_i + v_i$
$y^n \leftarrow \texttt{PolarEnc}(u^n)$
**return** $y^n$

---

The encoder input $\overline{p}_i$ refers to the subchannel parameter

$$\overline{p}_i = h_B^{-1}(H(U_i | U^{i-1}, X^n)). \tag{7}$$

This can be calculated offline using the techniques used to compute subchannel quality for communication described at the end of Section 2. The $\texttt{SoftPolarDec}(u^{i-1}, x^n)$ subroutine outputs

$$\Pr(U_i = 0 | U^{i-1} = u^{i-1}, X^n = x^n), \tag{8}$$

which can be calculated with $O(n \log n)$ complexity using a recursion given by Arikan [2009]. The `PolarEnc`$(u^n)$ subroutine simply multiplies by the generator matrix in: $y^n = u^n G_n$. This can be implemented in $O(n \log n)$ by exploiting the recursive structure, as noted earlier. The `Compress`$(\Delta^n, \bar{p}^n)$ and `Decompress`$(b, \bar{p}^n)$ routines can be any prefix-free lossless compressor/decompressor pair that uses at most

$$c + \sum_{i=1}^{n} \left[ \mathbf{1}(\Delta_i = 1) \log \frac{1}{1/2 - \bar{p}_i} + \mathbf{1}(\Delta_i = 0) \log \frac{1}{1/2 + \bar{p}_i} \right] \tag{9}$$

bits to send $\Delta^n$, where $c$ is some constant independent of $n$ and $\Delta^n$. Arithmetic coding (Rissanen [1976]) is a widely-used scheme that achieves this guarantee with $c = 2$.

This defines `PolarSim` for $n$ that is a power of two. An arbitrary $n$ can be handled by partitioning $\{1, \ldots, n\}$ into subsets, each of which has a cardinality that is a power of two. One then applies `PolarSim` to each subset separately. Since the coding is prefix-free, the encoder outputs can simply be concatenated together, as noted in Sec. 2.1.

### 3.3 Theoretical Guarantees

If one uses a linear complexity algorithm for `Compress` and `Decompress`, then the overall complexity of `PolarSim` is $O(n \log n)$ for both encoding an decoding. Using the fact that polar codes achieve capacity for channels satisfying the symmetry condition in (4), we can show that `PolarSim` is rate optimal in the large-$n$ limit, making it currently the only scheme with subexponential complexity in $n$ with a comparable guarantee.

**Theorem 1.** *Consider a joint distribution $P_{XY}$ in which $Y$ is binary and uniform and the reverse channel $P_{X|Y}$ satisfies the symmetry condition in (4). Suppose* `Compress` *and* `Decompress` *achieve the guarantee in (9).*

1. *(Correctness:) Algorithms 1 and 2 simulate the channel $P^{\times n}(Y|X)$ exactly: If $Z^n$ is i.i.d. $\mathrm{Unif}[0, 1]$, and $\bar{p}_i = h_B^{-1}(H(U_i|U^{i-1}, X^n))$, then the conditional probability that Algorithm 2 outputs $y^n$ given that $x^n$ is the input to Algorithm 1 is*

$$\prod_{i=1}^{n} P_{Y|X}(y_i|x_i). \tag{10}$$

2. *(Optimality:) Algorithms 1 and 2 are asymptotically rate optimal:*

$$\lim_{n \to \infty} \frac{1}{n} E[\ell(b)] \to I(X; Y), \tag{11}$$

*where $b$ is the output of the encoder.*

The proof is provided in Appendix D.

### 3.4 Experimental Results

We run `PolarSim` on the reverse $(P_{Y|X})$ version of three channels: (1) the BSC with a uniform input (2) the binary erasure channel, $X = Z \cdot Y$, where $Y$ is uniform over $\{-1, 1\}$ and $Z$ is Bernoulli($\epsilon$), and (3) the binary Gaussian channel $X = Y + Z$, where $Y$ is again uniform over $\{-1, 1\}$ and $Z$ is $\mathcal{N}(0, \sigma^2)$. Note that the reverse of the BSC with a uniform input is the BSC itself.

Fig. 3 and Fig. 4 show the rate performance of these simulations. Even at a block length of $2^{12}$, the performance is already close to the mutual information lower bound across all channels and rates. Performance improves with $n$ as expected, with both the average rate and the variance in the rate decreasing.

We also compare our scheme to the state-of-the-art scheme for channel simulation, Greedy Poisson Rejection Sampling (GPRS) [Flamich, 2024] (see Appendix E for the implementation details). Fig. 5 shows that `PolarSim` significantly outperforms GPRS in terms of the communication rate, even when the latter is optimized for the channel at hand. In Table 1, we can see that its computational efficiency is also significantly better, by several orders of magnitude. This is due to the exponential computational complexity of GPRS in $n$ compared to the pseudolinear complexity of `PolarSim`.

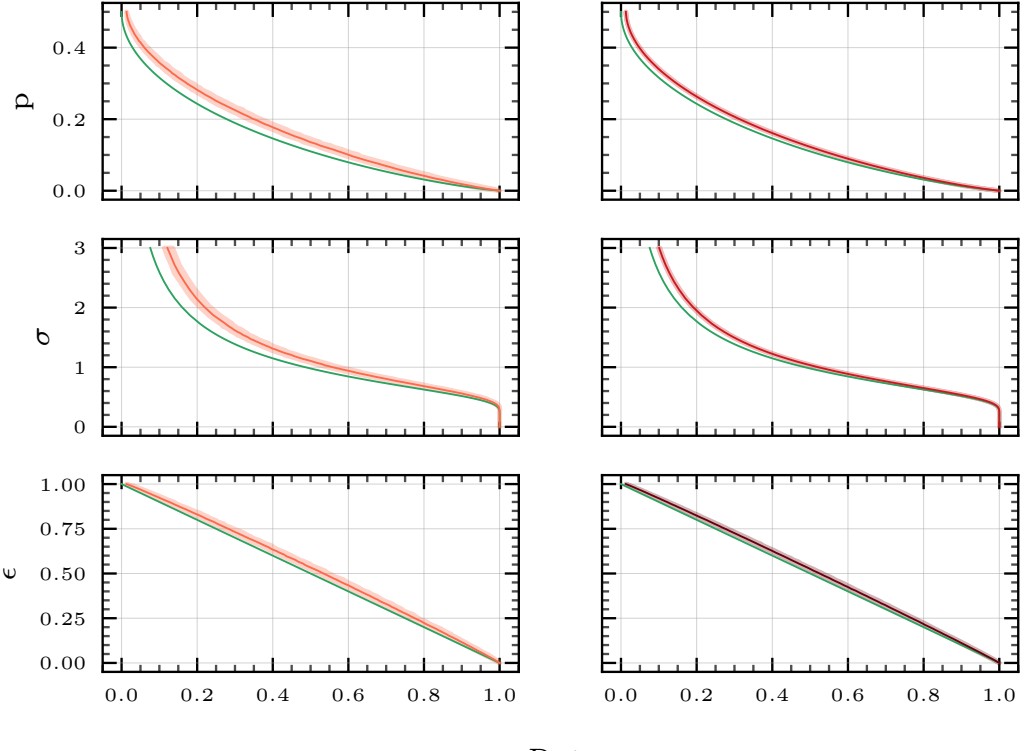

**Figure 3:** Rates achieved by `PolarSim` at different block lengths — ■ $n = 2^{12}$ (left), ■ $n = 2^{17}$ (top-right and middle-right), ■ $n = 2^{14}$ (bottom-right) for different noise levels across different channels, compared against the theoretical lower bound ■ $I(X;Y)$ . **Top:** $\mathsf{BSC}_p$ for $p \in (0, \frac{1}{2})$, **Middle:** Gaussian for $\sigma \in (0, 3)$, **Bottom:** Erasure for $\epsilon \in (0, 1)$. The lines represent the median values, and the boundaries of shaded regions represent the $5^{\text{th}}$ to $95^{\text{th}}$ percentile rates over 200 simulation runs.

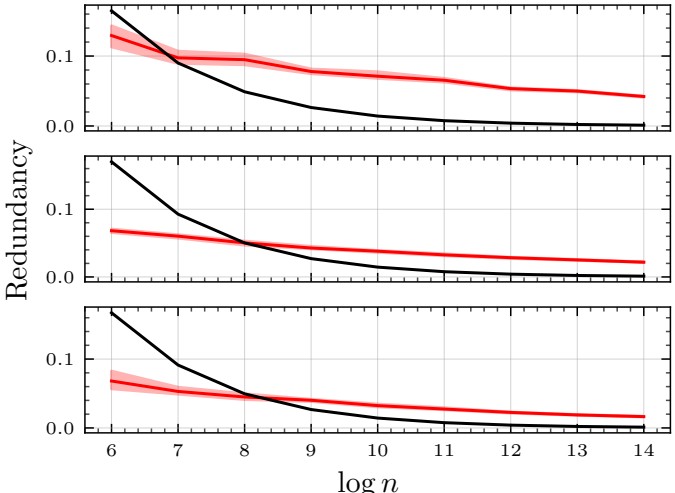

**Figure 4:** The redundancy of `PolarSim` is plotted for certain fixed channels (**Top:** BSC with $p = 0.05$, **Middle:** Reverse binary Gaussian channel with $\sigma = 0.5$, **Bottom:** Reverse binary erasure channel with $\epsilon = 0.2$) as the block length $n$ is varied. The plotted curve (■) is the median redundancy over 200 simulations, with the boundaries of the shaded region showing the bootstrapped 95% confidence interval around the sample median. The redundancy is defined as the gap between the achieved rate and the mutual information lower bound. For comparison, the theoretical maximum redundancy of PFRL (■) is also plotted for the respective channels (see 3 ). We see that for large block lengths, `PolarSim` has a higher redundancy, which is consistent with known results from channel coding [Mondelli et al., 2016].

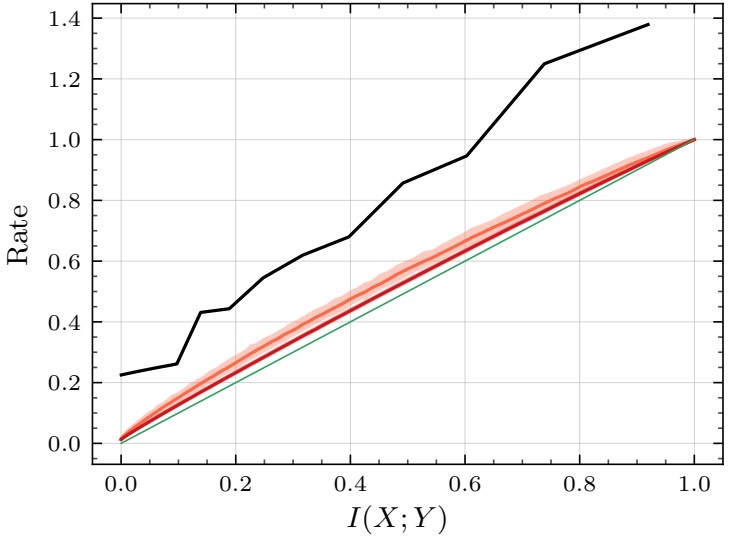

**Figure 5:** Comparison of schemes for BSC simulation: Average rates for `PolarSim` with ▧ $n = 2^{12}$ and ▧ $n = 2^{17}$ compared against ■ GPRS with $n = 8$ and the theoretical lower bound ■ $I(X; Y)$ over 1000 simulation runs.

| p | PolarSim | | | GPRS | | | $\lambda$ |
|---|---|---|---|---|---|---|---|
| | **Median** | **p5** | **p95** | **Median** | **p5** | **p95** | |
| 0.00 | 0.009 | 0.008 | 0.01 | 192.0 | 161.4 | 205.8 | 21333 |
| 0.25 | 0.009 | 0.008 | 0.01 | 246.1 | 180.7 | 415.7 | 27347 |
| 0.49 | 0.009 | 0.008 | 0.01 | 227.7 | 176.1 | 318.7 | 25305 |

**Table 1:** Execution time comparison between `PolarSim` and GPRS, for simulating BSC's with block length $n = 2^{12}$. The reported statistics are computed over 1000 trials for each value of the crossover probability $p$. GPRS cannot directly simulate such large block lengths. Therefore, the GPRS runtimes are obtained by scaling up the runtime for $n = 8$ blocks. This is justified by subadditivity (see (1)). The column $\lambda$ computes the ratio between the medians of the two schemes. For our chosen block length, `PolarSim` performs over four orders of magnitude faster than GPRS.

## 4 Related Work

### 4.1 Performance Bounds

Arguably the first to consider the channel simulation problem was Wyner [1975], who studied the problem without common randomness. The problem with common randomness was first introduced in the context of quantum information theory literature by Bennett et al. [2002] and Winter [2002]. Harsha et al. [2007] proposed a greedy one-shot channel simulation algorithm that achieved a logarithmic rate redundancy with respect to the lower bound for discrete channels. Li and El Gamal [2018] showed a similar achievability result that generalizes to more general channels using their strong Poisson functional representation lemma. Sriramu and Wagner [2024] showed via a two-stage rejection sampling scheme that an even lower redundancy was achievable in the i.i.d. case. Li and Anantharam [2021] extended the Poisson representation to obtain a generalised method for deriving one-shot achievability in different settings (see also Phan et al. [2024]).

### 4.2 Practical Schemes

The achievability proof in Li and El Gamal [2018] inspired several practical schemes that exploit properties of the Poisson process and perform well at short block lengths [Flamich et al., 2022, Flamich, 2024]. Similarly, the rejection sampler proposed by Harsha et al. [2007] has been generalized by Flamich et al. [2024] to work for arbitrary probability spaces. None of these schemes exhibit subexpoential complexity with $n$ when applied to product channels, however. If one restricts attention

to $n = 1$ and unimodal distributions, then improved schemes are possible [Flamich et al., 2024, Hegazy and Li, 2022], although by their nature such schemes cannot harness the amortization gain associated with increasing $n$.

Chou et al. [2018] addresses the problem of total variation approximate channel simulation and proposes a fixed rate scheme based on a soft covering argument that uses polar codes.

As noted in the introduction, the primary application of the channel simulation task is learned compression. Lei et al. [2024] and Li et al. [2020] consider how vector quantization methods can be directly applied to the compression task without explicitly simulating a channel. Although their methods do not simulate an i.i.d. channel, their use of ideas from error-correcting codes and vector quantization makes them the closest prior works to the present paper, along with Chou et al. [2018].

## 5   Concluding Remarks

`PolarSim` shows how polar codes can be used to simulate channels with favorable scalability in the dimension. By harnessing the attending amortization gain, the codes are able to realize significantly improved performance compared with existing schemes. Our focus has been on how the original form of polar codes can simulate symmetric, binary output channels. Subsequent extensions of polar codes [Şaşoğlu et al., 2009] could potentially be used to simulate arbitrary channels. This would be an interesting topic of future research.

The aim of the paper, however, is not to show that polar codes are useful for simulation *per se*. Rather, we seek to make the larger point that ideas from the field of error-correcting codes are useful for the simulation problem. We have used polar codes as an exemplar, but one could potentially apply turbo codes (Berrou et al. [1993]), low-density parity-check codes LDPCs (Gallager [1962]), algebraic codes [Blahut, 2003], trellis-coded modulation/quantization (Taubman et al. [2002]), sparse superposition codes (Joseph and Barron [2012]), or other codes (e.g., Caire et al. [1998]) instead. The `PolarSim` method is not expected to generalize to these other families. Polar codes are linear, however, and one can transform them into a simulator for the BSC using a different approach that relies on linearity alone. This is discussed in the supplementary materials, where we find that this alternate scheme achieves comparable performance to `PolarSim`. This alternate approach is directly applicable to other linear codes such as LDPCs.

In learned compression applications, the channels of interest are generally continuous. Error-correcting schemes that operate directly on continuous channels, such as trellis-coded modulation/quantization, are therefore of particular interest. To illustrate the potential of these codes, in the supplementary materials we show how trellis-coded quantization can be used to approximately simulate a Gaussian channel. The simulation is not exact, but in practical applications, exact simulation may be unnecessary.

## 6   Acknowledgement

This research was supported by the US National Science Foundation under grant CCF-2306278 and by a gift from Google. The authors wish to thank Lucas Theis and Jona Ballé for helpful discussions. They would also like to thank the anonymous reviewers for the feedback and suggestions they provided.

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

## A    Rate of the Toy Scheme

The rate of this scheme is governed by the cost of compressing the correction bit $\Delta$. We have

$$\Pr(\Delta = 1) = E\left[\Pr(\Delta = 1|X)\right] \tag{12}$$

$$= E\left[\frac{1}{2} - \min_{j \in \{0,1\}} p_{Y|X}(j|X)\right] \tag{13}$$

$$= \frac{1}{2} - E\left[\min_{j \in \{0,1\}} p_{Y|X}(j|X)\right] \tag{14}$$

$$\implies H(\Delta) = h_B\left(\frac{1}{2} - E\left[\min_{j \in \{0,1\}} p_{Y|X}(j|X)\right]\right). \tag{15}$$

Rearranging this, we can obtain

$$E\left[\min_{j \in \{0,1\}} p_{Y|X}(j|X)\right] = \frac{1}{2} - h_B^{-1}(H(\Delta)). \tag{16}$$

Next, consider

$$I(X;Y) = 1 - H(Y|X) \tag{17}$$

$$= 1 - E\left[h_B\left(\min_{j \in \{0,1\}} p_{Y|X}(j|X)\right)\right] \tag{18}$$

$$\geq 1 - h_B\left(E\left[\min_{j \in \{0,1\}} p_{Y|X}(j|X)\right]\right) \tag{19}$$

$$= 1 - h_B\left(\frac{1}{2} - h_B^{-1}(H(\Delta))\right). \tag{20}$$

Rearranging this, we finally obtain the required upper bound:

$$H(\Delta) \leq h_B\left(\frac{1}{2} - h_B^{-1}(1 - I(X;Y))\right). \tag{21}$$

## B    Trellis-Coded Quantization

The channel simulation scheme described in the the body of the paper is limited to binary-output channels. However, in many practical applications, we need to simulate channels with real-valued outputs. To demonstrate that the tools from coding theory are not limited to binary channels, we describe a scheme that uses *trellis coded quantization* for approximately simulating a AWGN channel for an i.i.d. Gaussian source.

The encoding algorithm for trellis coded quantization was originally developed as an efficient decoding algorithm for a family of channel coding schemes called *convolutional codes*. We will directly describe the construction used in source coding as it can be described in a self-contained manner.

A *fixed-rate time-invariant* trellis with $k$ states $\mathcal{S} = \{s_1, s_2, ..., s_k\}$ consists of a $k \times k$ binary transition matrix $G$ where each row and each column have exactly two nonzero entries, a set of disjoint codebooks $\mathcal{V} = \{\mathcal{C}_1, \mathcal{C}_2, ..., \mathcal{C}_m\}$ with each containing $2^{R-1}$ codewords from the source alphabet $\mathcal{X}$, and a map $T : \{1, \ldots, k\} \times \{1, \ldots, k\} \mapsto \mathcal{V}$ which assigns a codebook to each pair of states.

A path of length $n$ through the trellis consists of a *traversable* sequence of states $s_t = (s_{t_0}, s_{t_1}, ..., s_{t_n})$—i.e, all sequences such that $G(s_{t_i}, s_{t_{i+1}}) = 1$. For any sequence of codewords $\hat{x}^n$, we will use the shorthand $\hat{x}^n \in s_t$ if $\hat{x}^n$ can be produced by the transitions of the path—i.e., if $\hat{x}_i \in T(s_{t_{i-1}}, s_{t_i})$ for all $i$.

In a data compression setting, the encoder, upon receiving a source sequence $x^n$, examines all traversable paths through the trellis to find a codeword sequence $\phi(x^n)$ that is closest to $x^n$ under the mean squared error distortion metric:

$$\phi(x^n) = \operatorname*{arg\,min}_{\substack{\hat{x}^n \in s_0^n, \\ s_0^n \text{ is traversable}}} \|x^n - \hat{x}^n\|^2. \tag{22}$$

It then encodes this into a bit-string by specifying the sequence of state transitions used and the codeword index in the associated codebook for each transition. Our chosen structure for the transition matrix $G$ ensures that each transition can be encoded by one bit. Each codeword can be then indexed within the corresponding codebook using $R - 1$ bits, bringing the total communication rate to $R$ bits per symbol.

Although the search space for finding the best codeword grows exponentially in $n$, it can be implemented efficiently using the *Viterbi algorithm* [Viterbi, 1967]. Here, at every stage $k$ of our scheme, we keep track of the best paths terminating at every state $s$ and their associated costs recursively:

$$J(s,k) = \min_{\hat{s} \in \mathcal{S}} \left( J(\hat{s}, k-1) + \min_{\hat{x} \in T(\hat{s},s)} (x_k - \hat{x})^2 \right), \text{and} \tag{23}$$

$$U(s,k) = \arg\min_{\hat{s} \in \mathcal{S}} J(s,k). \tag{24}$$

The base case of the recursion is specified by the choice of the initial state $s_{t_0}$, which is typically set to be $s_1$. The initial costs are then set to be $J(s_1, 0) = 0$, and $J(\hat{s}, 0) = \infty$ for all $s_1 \neq \hat{s}$. The optimal state transition sequence and the corresponding codeword sequence $\phi(x_1^n)$ can be recovered using $U(\cdot, \cdot)$.

To encode a sequence of length $n$, the Viterbi algorithm performs $nm2^{R-1}$ single-letter squared distance evaluations and $2nk$ comparisons — thus, the computational cost grows only linearly in the block length making the algorithm highly scalable. Many practical data compression schemes use trellis codes due to this scalability, and because of their performance in terms of rate.

## B.1 Approximate AWGN Simulation Using Trellis-Coded Quantization

1. Let $X^n \sim \mathcal{N}(0, \sigma^2)$ be the input at the encoder, and $f^*(\cdot, R)$ be a rate $R$ trellis coded quantizer that obtains an average distortion value of $D$. We use a 256-state trellis with each state having exactly two branches leaving it, along with a codebook of size $2^{R+1}$ which is partitioned into 4 subsets, each of size $2^{R-1}$. Each branch of the trellis is then associated with one of the 4 subsets (See Figure 3.15 and Table 3.5 of Taubman et al. [2002] for a full description of the mapping used for this assignment ).

   The above Trellis is initialised with randomly generated codewords from a standard normal and is subsequently trained using the Lloyd-Max algorithm.

   Refer to Taubman et al. [2002] for a detailed description of the Trellis construction and training.

2. Using the common randomness, select a uniformly random rotation matrix $\Pi$ at both the encoder and decoder.

3. At the encoder, compute the randomly rotated source $\tilde{X}^n = \Pi X^n$ and the reconstruction $\hat{\tilde{X}}^n = f(\tilde{X}^n, R)$. Transmit the reconstruction to the decoder by specifying the corresponding trellis path.

4. Compute the scaling factor $a = \frac{\sigma^2}{\sigma^2 - D}$ and the reconstruction $\tilde{Y}^n = a\hat{\tilde{X}}^n$

5. Finally, compute $Y^n = \Pi^{-1}\tilde{Y}^n$ as the output of the channel simulation scheme.

   In figure 6, we generate 100 independent realizations of $X^n$ with $n = 1000$ and source power $\sigma^2 = 1$. The scheme outlined above is then used to generate the corresponding $Y^n$ realizations. The quantiles of the sample noise $D = \|X^n - Y^n\|^2$ are plotted against the theoretical quantiles obtained by assuming $p_{Y|X}$ to be AWGN with noise power $\frac{1}{n} \sum_{i=1}^{n} \|X_i^n - Y_i^n\|^2$.

   In Figure 7, we plot the rate of our approximate scheme (i.e. the rate of the trellis coded quantizer used) against the mutual information of the simulated channel. Bootstapped 95%ile confidence intervals for the mutual information are plotted to account for the error in estimating the noise power.

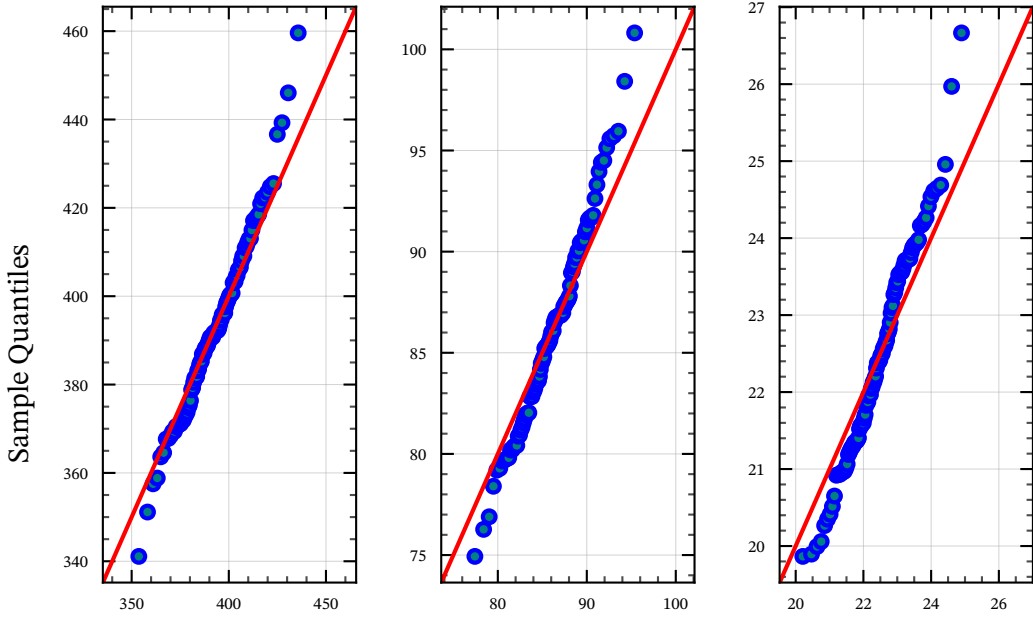

**Figure 6:** For trellis coded quantizers with rates $R = 1$ (**Left**), $R = 2$ (**Middle**) and $R = 3$ (**Right**), the quantiles for $M = 100$ realizations of the sample noise $D = \|Y^n - X^n\|^2$ are plotted against the theoretical quantiles obtained from an AWGN with noise power $\frac{1}{n} \mathrm{E}[D]$. We use the sample mean to estimate $\mathrm{E}[D]$.

Using a trellis with a fixed codebook limits the output alphabet to a finite set of real numbers. To simulate a continuous channel, we introduce a random orthogonal transformation, chosen by the common randomness, that is applied to the input before encoding. The decoder subsequently applies the inverse transformation to the reconstruction. This has the effect of randomizing the output symbols while retaining their proximity to the source on average.

We present the results of this scheme in Figures 6 and 7. We see that the distribution of the realizations of the simulated channel noise are fairly close to an AWGN with matching power, and that each of the operating rates of the trellis is very close to the mutual information of the simulated channel for that rate.

## C    Channel Simulation Using Arbitrary Linear Codes

Given any lattice $\Lambda \subset \mathbb{R}^n$, an associated quantizer $Q_\Lambda$, and a source realization $X^n$ observed at the encoder, Zamir and Feder [1992] provide a mechanism for simulating an additive noise channel whose noise distribution is uniform over the decoding cell of the lattice,

$$Q_\Lambda^{-1}(0^n) = \{z^n \in \overset{n}{\mathbb{R}} : Q_\Lambda(z^n) = 0^n\}, \tag{25}$$

using *dithered quantization*: we generate $Z^n \sim \mathrm{Unif}(Q_\Lambda^{-1}(0^n))$ using the common randomness, and at the encoder compute $Q_\Lambda(X^n + Z^n)$, which is transmitted to the decoder. Then, according to Theorem 1 in Zamir and Feder [1992], computing

$$Y^n = Q_\Lambda(X^n + Z^n) - Z^n \tag{26}$$

recovers a sample from the target distribution. An analogous fact holds for linear codes over finite fields.

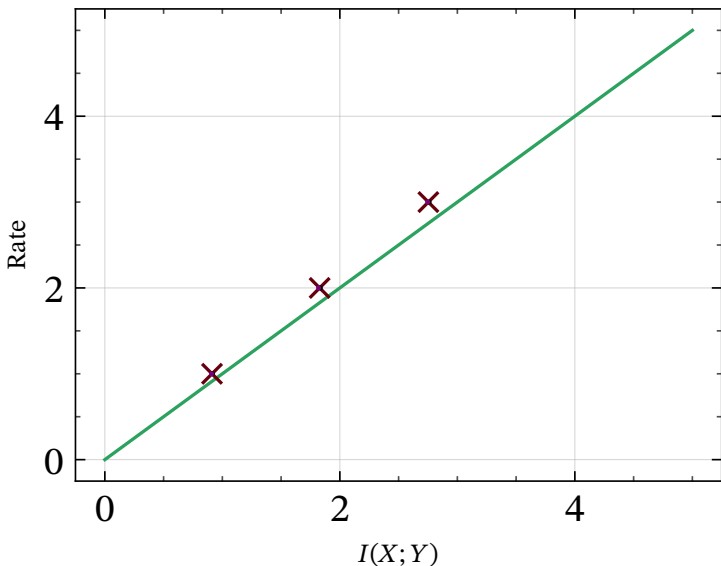

**Figure 7:** The rate of the trellis quantizer is plotted against the mutual information of the simulated channel. Bootstrapped 95% confidence intervals are plotted for the noise mean estimate.

### C.1  Alternative Scheme for the BSC Simulation Using Polar codes

For a linear code to be capacity-achieving over a channel, its decoding cell must have a certain structure. For binary codes to achieve the capacity of a BSC (such as polar codes) their decoding cells must resemble Hamming balls. Thus applying the above procedure with a polar code simulates an additive noise channel in which the noise is approximately uniformly distributed over a Hamming ball. Instead of using an additive dither to randomize the input to a deterministic quantizer, we can randomly shift the codewords to a coset of the original lattice to have the same effect. This is especially convenient for polar codes because shifting their cosets can be accomplished by randomizing the frozen bits, saving a call to the quantization routine.

Suppose we seek to simulate $n$ uses of a BSC with crossover probability $p$. The number of message bits is calculated as:

$$m = n \cdot (1 - h_B(p)) \tag{27}$$

and the number of frozen bits is $n - m$, which are generated using the common randomness. The simulation encoder takes the input bit string $X^n$ and the block of frozen bits. The simulation encoder runs the channel decoding algorithm with channel output $X^n$ and the given frozen bits to obtain $m$ message bits. These bits are passed to the simulation decoder, which runs the channel encoder to produce $Y^n$ using the received message bits and the frozen bits. The role of the encoder and decoder as thus swapped compared to the communication task, as was the case with `PolarSim` (see Fig. 8). The rate of this scheme is $m/n$.

This scheme does not result in an exact simulation of $n$ copies of the $\mathrm{BSC}_p$, however, in that $X^n \oplus Y^n$ is uniformly distributed over the cell rather than being i.i.d. Bernoulli($p$). This is both because an i.i.d. Bernoulli($p$) string is not constrained to a single Hamming ball and because the cell of the polar code is not exactly a Hamming ball to begin with. Fig 9 shows a histogram of the number of bit-flips in this scheme compared with the target binomial distribution. We see that the actual distribution of bit flips (top) has a mean that is too high (because the cell is not a true Hamming ball) and a variance that is too low (because the actual noise process is not constrained to a single Hamming ball).

But we can upgrade this simulator to one that achieves exact simulation using a pre- and post-correction process. The pre-correction is to adjust the parameter of the channel simulator to be $\hat{p} < p$, which is chosen empirically to result in the correct ($pn$) number of bits being flipped on average. The post-correction scheme corrects for deviation in the variance and any remaining discrepancy in the mean and operates as follows. The encoder uses its private randomness to generate a realization of a random variable $N$ that is binomially distributed with parameters $n$ and $p$. It then runs the above

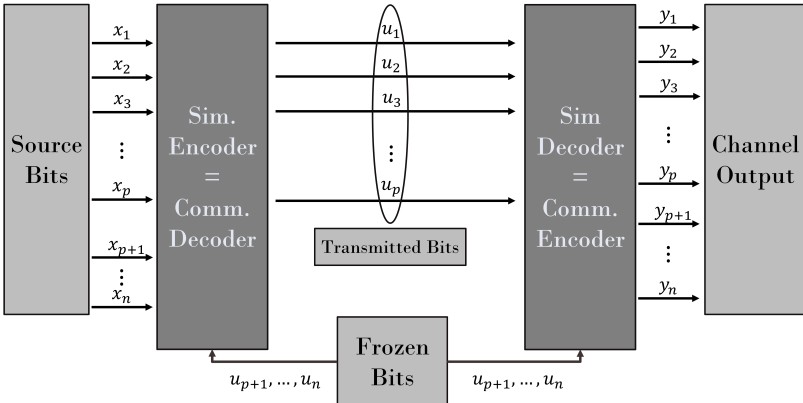

**Figure 8:** Block diagram representation of a BSC simulator based on polar codes. The role of the encoder and decoder are swapped compared with conventional communication.

process, with pre-correction, and computes the number of bits flipped by the scheme. It then sends a *correction message* to the decoder to ensure that the actual number of bits flipped is exactly $N$. The correction message takes two forms, depending on whether one is operating in the high-rate or low-rate regime:

1. If $m < \frac{n}{2}$: the encoder identifies the smallest $L$ so that the first $L$ bits of $Y^n$ can be overwritten in such a way that the resulting $Y^n$ string differs from $X^n$ in exactly $N$ positions. The encoder sends the decoder $L$ using $\log n$ bits and then the $L$ bits with which the decoder should overwrite the prefix of $Y^n$.

2. If $m \geq \frac{n}{2}$: the encoder identifies a set of $L$ bits that can be flipped to result in there being exactly $N$ flipped bits overall. The encoder communicates to the decoder a binary string of length $n$ that contains $L$ 1's and $n - L$ zeros. The encoder communicates $L$ to the decocder using $\log n$ bits. It then communicates the bit string using $2 + \log \binom{n}{L}$ bits, which can be realized using arithmetic coding.

By construction, the post-correction scheme ensures that the noise sequence $Z^n = X^n \oplus Y^n$ has the correct distribution. The simulator is still not exact, however, because the distribution of $Z^n$ is not exchangeable. This can be rectified by generating a random permutation of $\{1, \ldots, n\}$, say $\Pi$, from the common randomness and having the encoder apply $\Pi$ to $X^n$ at the start and having the decoder apply $\Pi^{-1}$ to $Y^n$ at the end. Note that an exchangeable distribution over binary strings with the property that the number of ones has a binomial distribution must be i.i.d. Bernoulli.

Note that the pre-correction part of the scheme is not needed to ensure exact simulation; it only reduces the burden on the post-correction scheme. Also note that both the high-rate and low-rate schemes can be used for any value of $m$. We found empirically that the above division performed better than relying on either one alone. The performance of the corrected scheme is shown in (Fig. 10), which shows that the performance is similar to `PolarSim`. The corrected scheme can never perform better than the uncorrected scheme, but the difference between the two vanishes as $n$ increases.

The idea of fixing the number of bit flips using the common randomness is reminiscent of the two-stage channel simulation scheme of Sriramu and Wagner [2024] and the layered randomized quantizer of Hegazy and Li [2022], which are without loss of optimality in their respective setups. Note that the above scheme can be used to simulate a BSC using any linear code if one uses the dithered approach in (26) instead of relying on frozen bits.

## D   Proof of Theorem 1

**Theorem 1.** *Consider a joint distribution $P_{XY}$ in which $Y$ is binary and uniform and the reverse channel $P_{X|Y}$ satisfies the symmetry condition in (4). Suppose* `Compress` *and* `Decompress` *achieve the guarantee in (9).*

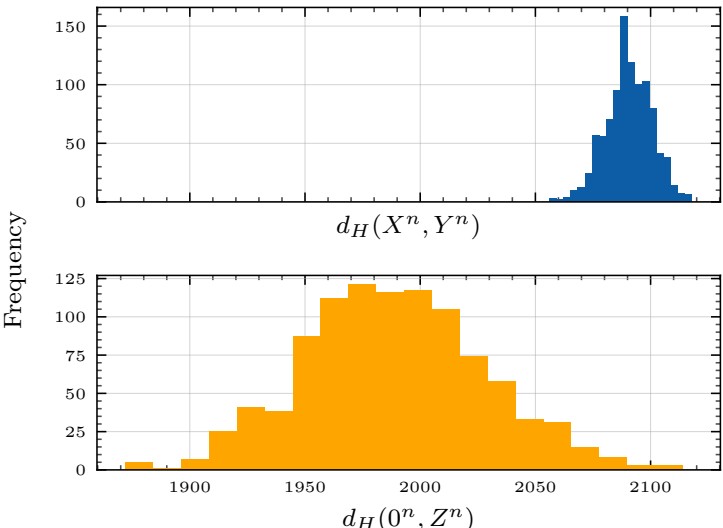

**Figure 9: Top**: The distribution of the hamming distance, $d_H(X^n, Y^n) = \sum_{i=1}^{n} X_i \oplus Y_i$, between the input binary string $X^n$ and the output $Y^n$ produced by the uncorrected polar simulator. **Bottom**: The target distribution, with each $Z_i \sim \text{Bern}(p)$ i.i.d..

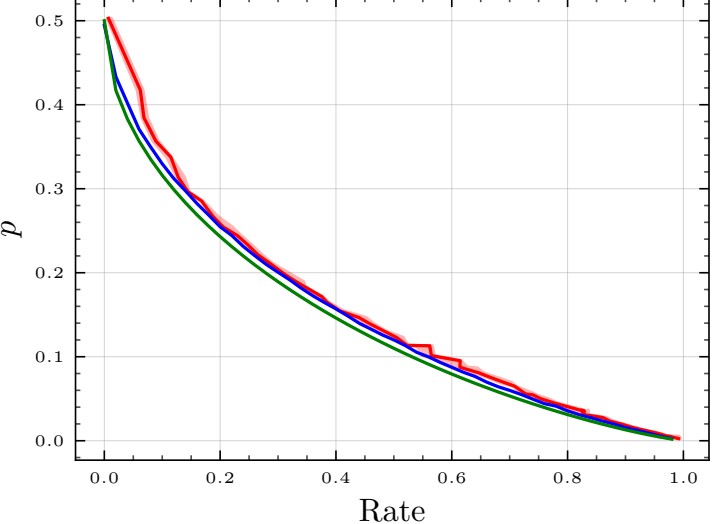

**Figure 10:** BSC simulation using the polar code simulator with post-correction. The median post-correction rate (■) is plotted along with the uncorrected rate (■) and the mutual information lower bound (■). The boundaries of the shaded show the maximum and minimum rate for the corrected scheme over five simulation runs.

1. *(Correctness:) Algorithms 1 and 2 simulate the channel $P^{\times n}(Y|X)$ exactly: If $Z^n$ is i.i.d. $\mathrm{Unif}[0,1]$, and $\bar{p}_i = h_B^{-1}(H(U_i|U^{i-1}, X^n))$, then the conditional probability that Algorithm 2 outputs $y^n$ given that $x^n$ is the input to Algorithm 1 is*

$$\prod_{i=1}^{n} P_{Y|X}(y_i|x_i). \tag{28}$$

2. *(Optimality:) Algorithms 1 and 2 are asymptotically rate optimal:*

$$\lim_{n \to \infty} \frac{1}{n} E[\ell(b)] \to I(X;Y), \tag{29}$$

*where b is the output of the encoder.*

*Proof.* Suppose $n$ is a power of two, and consider the following joint distribution between $(U^n, X^n, Y^n)$:

$$\Pr(U^n = u^n, Y^n = y^n, X^n = x^n) = \prod_{i=1}^{n} P_X(x_i) \prod_{i=1}^{n} P_{Y|X}(y_i|x_i) 1(u^n = y^n G^{-1}) \tag{30}$$

$$= \frac{1}{2^n} 1(y^n = u^n G) \prod_{i=1}^{n} P_{X|Y}(x_i|y_i). \tag{31}$$

Given the input $X^n$, the $U^n$ string generated by the encoder has distribution $P_{U|X}^{\times n}$ by construction, since $\texttt{SoftPolarDec}(x^n, u^{i-1}, P_{X|Y})$ computes $\Pr(U_i = 0|X^n = x^n, U^{i-1} = u^{i-1})$ under the distribution in (30)-(31). Due to the lossless nature of the compression, $\Delta^n$ is recovered exactly at the decoder. Since $V^n$ is common to the two terminals, $U^n$ is thus recovered exactly by the decoder. Then setting $Y^n = U^n \cdot G$ results in $(X^n, Y^n)$ having the joint distribution $P_{XY}^{\times n}$ as desired. This establishes the correctness of the algorithm for $n$ that is a power of two. Correctness of the algorithm for any $n$ immediately follows.

Turning to optimality, again suppose $n$ is a power of two. For the sequence $\Delta^n$ we have

$$\Pr(\Delta_i = 1) = \frac{1}{2} - \sum_{u^{i-1}, x^n} P_{U^{i-1}, X^n}(u^{i-1}, x^n) \min\Big( \Pr(U_i = 1|U^{i-1} = u^{i-1}, X^n = x^n), \tag{32}$$

$$\Pr(U_i = 0|U^{i-1} = u^{i-1}, X^n = x^n)\Big) \tag{33}$$

$$= \frac{1}{2} - \sum_{u^{i-1}, x^n} P_{U^{i-1}, X^n}(u^{i-1}, x^n) h_B^{-1}(H(U_i|U^{i-1} = u^{i-1}, X^n = x^n)) \tag{34}$$

$$\leq \frac{1}{2} - h_B^{-1}(H(U_i|U^{i-1}, X^n)) \tag{35}$$

$$= \frac{1}{2} - \bar{p}_i, \tag{36}$$

where the inequality follows by the convexity of $h_B^{-1}(\cdot)$. Thus the average rate may be bounded as

$$\frac{1}{n} E[\ell(b)] \leq \frac{1}{n} \sum_{i=1}^{n} \left[ \Pr(\Delta_i = 1) \log \frac{1}{1/2 - \bar{p}_i} + \Pr(\Delta_i = 0) \log \frac{1}{1/2 + \bar{p}_i} \right] + \frac{c}{n} \tag{37}$$

$$\leq \frac{1}{n} \sum_{i=1}^{n} h_B\left(\frac{1}{2} - \bar{p}_i\right) + \frac{c}{n}. \tag{38}$$

We bound this quantity using the polarization property. Fix $\delta > 0$. By Arikan [2009], for all sufficiently large $n$ there is a set $\mathcal{N}_n$ of "noisy" indices such that

$$\frac{|\mathcal{N}_n|}{n} \geq 1 - I(X;Y) - \delta \tag{39}$$

$$h_B(1/2 - \bar{p}_i) \leq \delta \quad \text{for all } i \in \mathcal{N}_n. \tag{40}$$

The rate is thus upper bounded by

$$\frac{\mathrm{E}[\ell(b)]}{n} \leq \frac{n - |\mathcal{N}_n|}{n} + \frac{1}{n} \sum_{i \in \mathcal{N}_n} h_B(1/2 - \bar{p}_i) + \frac{c}{n} \tag{41}$$

$$\leq \frac{n - |\mathcal{N}_n|}{n} + \frac{\delta \cdot |\mathcal{N}_n|}{n} + \frac{c}{n} \tag{42}$$

$$\leq I(X;Y) + 2\delta + \frac{c}{n}. \tag{43}$$

If we let $\bar{R}_n = \mathrm{E}[\ell(b)]/n$ denote the minimum rate of `PolarSim` at block length $n$ (minimized over all partitions if it is not a power of two), then $n\bar{R}_n$ is itself subadditive. As $n$ increases through powers of two, from (43) we have

$$\bar{R}_n \to I(X;Y). \tag{44}$$

and thus

$$\inf_n \bar{R}_n = I(X;Y). \tag{45}$$

The result then follows by subadditivity (cf. 1). $\qquad\square$

## E   GPRS Implementation Details

We implement Algorithm 3 from Flamich [2024]. The proposal distribution $P$ is chosen to be i.i.d. Bernoulli(1/2) with $n = 8$. Given the input $X^n = x^n$, the target distribution $Q(y^n)$ is chosen to be $\prod_{i=1}^n \mathsf{BSC}_p(y_i|x_i)$, where $p$ ranges over $(0, 1/2)$. The stretch function $\sigma$ was derived using the definitions provided in [Flamich 2023]:

$$w_P(h) = F\left(\frac{\log h - n(\log(1-p) + 1)}{\log p - \log(1-p)}, n, \frac{1}{2}\right),$$

$$w_Q(h) = F\left(\frac{\log h - n(\log(1-p) + 1)}{\log p - \log(1-p)}, n, p\right), \text{ and}$$

$$\sigma(h) = \int_0^h \frac{1}{w_Q(\eta) - \eta w_P(\eta)} d\eta,$$

where $F(\cdot, n, p)$ is the CDF of a Binomial$(n, p)$ random variable. This stretch function was evaluated numerically for each input.

The algorithm outputs a positive integer $n$, which is entropy coded using the Zeta distribution in [Flamich 2023, (151)]. The number of bits is divided by $n = 8$ to obtain the rate. For each $p$ on a grid in $(0, \frac{1}{2})$, we run the simulation 200 times and plot the average rate obtained against the channel mutual information $1 - H(p)$.

We now make use of the fact that the selection rule employed by the algorithm needs to be evaluated for each point in the output sample space only once — if the first occurrence of an output sequence in the randomly generated codebook is rejected, all its subsequent occurrences are also rejected. This simplification helps speed up the execution significantly and also reduces the rate as repetitions need not be indexed.

