# OpenReview forum: "Fast Channel Simulation via Error-Correcting Codes"
_NeurIPS.cc/2024/Conference — NeurIPS 2024 poster_

### Official Review · Reviewer_w9ei · 2024-06-14

**Soundness:** 3
**Presentation:** 2
**Contribution:** 3
**Rating:** 7
**Confidence:** 4

**Summary:**

Inspired by the duality between source and channel coding, the authors use polar codes to develop a channel simulation algorithm for binary output channels. Notably, the authors' scheme scales as $O(n \log n)$ where $n$ is the channel dimension, providing an example of a channel simulation algorithm whose runtime scales sub-exponentially in $n$ that doesn't rely on dithered quantization. The authors conduct some toy experiments to showcase the behaviour of their algorithm.

**Strengths:**

Overall, I am very excited about this paper. The general idea of exploiting the duality between source and channel coding to develop fast channel simulation algorithms could significantly impact the field and thus have far-reaching implications for neural compression.

The authors' concrete scheme is relatively simple, leaving little doubt about its correctness and efficiency. However, they do back things up with appropriate theoretical results. I have checked the proofs of the statements and can confirm that they are correct.

Finally, the authors provide further discussion in Appendices A and B on possible future directions / concrete examples of channel codes that could be used to develop new channel simulation algorithms.

**Weaknesses:**

I should note that while I have expertise with channel simulation algorithms and have some basic familiarity with channel codes, I am not an expert in the latter.

Given the above, I found the paper's biggest high-level weakness to be its non-replicability, though most of it should be easily fixable. The biggest issue is that the description of the experimental setup for the comparison study between PolarSim, GPRS and PFR is missing, which means that Figure 3 and the last paragraph in the section are not interpretable. While the first set of experiments in Section 3.2 is better explained, the authors should also include the analytic form of the mutual information (the green curves in Figure 2) to make the results more interpretable. Similarly, the contents of Appendix A are quite high-level, with most of the experimental details missing; hence, again, Figure 4 is not interpretable. These problems should be easily fixable by providing the essential details of the experiments in the main text and an additional section in the appendix that describes the precise setup; I am happy to increase my score once the authors address this. I saw that the authors provided their code in the supplementary material; hence, the paper's contents are technically reproducible. Still, ideally, the paper should have sufficient detail so that someone can reproduce the experiments without the authors' code.

The second weakness of the paper is that it is closer to a position paper both in terms of content and impact. As the authors explain, "The aim of the paper, however, is not to show that polar codes are useful for simulation per se. Rather, we seek to make the larger point that ideas from the field of error-correcting codes are useful for the simulation problem." Now, I believe the paper should be accepted just based on the strength of the idea (given that the authors address my first concern). However, the paper could have been significantly stronger if the authors considered some practical applications of their scheme and carried out more extensive experiments to demonstrate its usefulness.

## Typos & Miscellaneous:
 - line 89: should be $h_B^{-1}(h_B(p)) =  p$
 - line 99: "$m + n$y" instead of $m + n$.
 - line 120: $F^{\otimes n}$ undefined - I believe the authors mean Kroneker product
 - line 125: "thatl"
 - line 231: "mutual information lower " - the word "bound" is missing
 - line 479: incorrect reference to figure 7, should be figure 8; the contents of figure 7 are never referenced
 - lines 502-503: "has the correct binomial distribution." - I believe it should be bernoulli
 - Eqs (21) and (22): Indicator has $Z_i$ instead of $Z_1$ and $Z_2$
 - Figs 1a & b are difficult to read; please increase the font size of the axis, tick and legend labels
 - I think the left and right panels of Figure 2 could be merged
 - Please add an explicit reference to the proof of Theorem 1 in Appendix C.
 - Eq (28): I believe the expression should be $I[U_2 ; Y_1, Y_2, \mid U_1]$, unless $U_1$ is independent of $U_2$. Given Eq (17), it makes sense, but the authors should mention this explicitly.
 - Regarding the proof of Theorem 1 in Appendix C:
   - Please restate the theorem in the appendix.
   - Why does eq 53 mix the two probability notations?
   - I believe Eq (54) is an equality rather than an inequality.

**Questions:**

Not so much a question but a suggestion: In Figure 3, the authors note that "Data for GRPS is only plotted for parameters where the algorithm consistently terminates in a reasonable amount of time." If I understand correctly, the authors are simulating a vector of 8 iid Bernoulli channels. Since, in this case, the sample space is finite and has few elements (only 256), the implementations of PFR and GPRS can be simplified. To see this, note that if we draw the same sample twice, the one with the later arrival time will never be accepted. This motivates a simple strategy: simulate the arrival time for each element of the sample space and perform a brute-force search to find the element that fulfils the acceptance criterion. In the case of PFR, this is equivalent to just performing the Gumbel-max trick. It would be good if the authors could improve their implementation using the above solution (or otherwise) to provide a more complete comparison of the methods in Figure 3.

**Limitations:**

While the authors clearly state that polarization occurs for large $n$, providing a plot of this phenomenon would be valuable. For a fixed amount of mutual information (e.g., uniformly distributed across the dimensions), the authors could include a plot of the rate of PolarSim versus the problem dimensionality.

---

> ### Author Rebuttal · Authors · 2024-08-07
>
> Thank you for your encouraging feedback, and for suggesting the improvement to our comparison plot.
> > ...  description of the experimental setup ...
>
> We will describe the experimental setup for Figure 3 here, which we will also include in the camera-ready version of the paper. This also takes into account the reviewer's suggestion for speeding up the implementation of GPRS and PFR. The updated plot is included in the attached PDF (Please refer to Figure 2).
> 1. $\mathtt{PolarSim}$: We use the same setup used in the BSC plots for Figure 2 (i.e. the top plots). We plot the average rate over 200 simulation runs.
> 2. **GPRS**: We implement [7, Alg. 3]. The proposal distribution $P$ is chosen to be i.i.d. Bernoulli($1/2$) with $n = 8$. Given the input $X^n = x^n$, the target distribution $Q(y^n)$ is chosen to be $\prod_{i = 1}^n \mathsf{BSC}_p(y_i|x_i)$, where $p$ ranges over $(0,1/2)$. The stretch function $\sigma$ was derived using the definitions provided in [7]: $w_P(h) = F\left(\frac{\log h - n( \log(1-p) + 1 )}{\log p - \log(1 - p)}, n, \frac{1}{2}\right)$, $w_Q(h) = F\left(\frac{\log h - n( \log(1-p) + 1 )}{\log p -\log(1 - p)}, n, p\right)$, and $\sigma(h) = \int_0^h \frac{1}{w_Q(\eta) - \eta w_P(\eta)}d\eta$ where $F(\cdot,n,p)$ is the CDF of a $\text{Binomial}(n,p)$ random variable.
> The algorithm outputs a positive integer $n$, which is entropy coded using the Zeta distribution in [(151), 7]. The number of bits is divided by $n = 8$ to obtain the rate. For each $p$ on a grid in $(0, \frac{1}{2})$, we run the simulation $200$ times and plot the average rate obtained against the channel mutual information $1 - H(p)$. We observe that the selection rule used by the algorithm can overlook repetitions --- if the first occurrence of an output sequence in the randomly generated codebook is rejected, all its subsequent occurrences are also rejected. This simplification helps speed up the execution significantly and also reduces the rate as repetitions need not be indexed.
>
> 3. **PFR**: We use the algorithm described in  [Sections II-III, 1] with $P_Y$ chosen to be i.i.d. Bernoulli($1/2$)
>     with $n = 8$ and $P_{Y|X}$ chosen to be i.i.d. BSCs with crossover probability $p$. We use the same idea as outlined before in the GPRS implementation to eliminate repetitions from the codebook, speeding up the algorithm and improving the rate. The selected index $n$ is compressed using the Zipf distribution given in [Sec. III, 1]. We simulate the channel $200$ times for uniformly randomly chosen input sequences and plot the average rate obtained.
>
> > ... contents of Appendix A ...
>
> Proposed algorithm:
> 1. Let $X^n \sim \mathcal{N}(0,\sigma^2)$ be the input at the encoder, and $f^*(\cdot, R)$ be a rate $R$ trellis coded quantizer that obtains an average distortion value of $D$. We use a $256$-state trellis with each state having exactly two branches leaving it, along with a codebook of size $2^{R+1}$ which is partitioned into 4 subsets, each of size $2^{R-1}$. Each branch of the trellis is then associated with one of the $4$ subsets (See [Figure 3.15, 14] for the mapping used). The Trellis is initialized with randomly generated codewords from a standard normal and trained using the Lloyd-Max algorithm. Please refer to [14] for a detailed description of the Trellis construction and training.
>
> 2. Using the common randomness, select a uniformly random rotation matrix $\Pi$ at both the encoder and decoder.
>
> 3. At the encoder, compute the randomly rotated source $\tilde{X}^n = \Pi X^n$ and the reconstruction $\hat{\tilde{X}}^n = f(\tilde{X}^n, R)$. Transmit the reconstruction to the decoder by specifying its trellis path.
> 4. Compute the scaling factor $a = \frac{\sigma^2}{\sigma^2 - D}$ and the reconstruction $\tilde{Y}^n = a\hat{\tilde{X}}^n$
> 5. Finally, compute $Y^n = \Pi^{-1}\tilde{Y}^n$ as the output of the scheme.
>
> In figure 4, we generate $100$ independent realizations of $X^n$ with $n=1000$ and source power $\sigma^2 = 1$. The scheme outlined above is then used to generate the corresponding $Y^n$ realizations. The quantiles of the sample noise $D = \|| X^n - Y^n \||^2$ are plotted against the theoretical quantiles obtained by assuming $p_{Y|X}$ to be AWGN with noise power $\frac{1}{n}\sum\limits_{i=1}^{n}\|| X_i^n - Y_i^n \||^2$.
>
> In Figure 5, we plot the rate of our approximate scheme against the mutual information of the simulated channel. Bootstrapped $95\%$ile confidence intervals for the mutual information are plotted to account for the error in estimating the noise power.
> > ... analytic form of the mutual information ...
>
> 1. $p_{Y|X}$ is $\mathsf{BSC(p)}$, $p_X$ is $\text{Unif}(\{0,1\})$}: $I(X;Y) = 1 - H(p)$
> 2. $p_{Y|X}$ is $\mathsf{BEC(\epsilon)}$, $p_X$ is $\text{Unif}(\{0,1\})$}: $I(X;Y) = 1 - \epsilon$
> 3. $p_{Y|X}$ is $\mathsf{AWGN(\sigma^2)}$, $p_X$ is $\text{Unif}(\{-1,1\})$}: There is no known closed-form expression for the mutual information in this case. We note that $p_Y(y) = \frac{1}{2}\mathcal{N}(y|-1, \sigma^2) + \frac{1}{2}\mathcal{N}(y|1, \sigma^2)$, $p_{Y|X}(y|x = 1) = \mathcal{N}(y|1, \sigma^2)$, and $p_{Y|X}(y|x = -1) = \mathcal{N}(y|-1, \sigma^2)$. We can then calculate $h(Y)$ using numerical integration and observe that $h(Y|X) = \frac{1}{2}h(Y|X=1) + \frac{1}{2}h(Y|X=-1) = \frac{\ln 2}{2} \left[ 1 + \ln{ 2\sigma^2\pi } \right]$. From these, we can compute $I(X;Y) = h(Y) - h(Y|X)$.
>
> >...polarization occurs for large $n$, ..
>
> To clarify, the claim in the paper was that, for a fixed channel, polarization increases with the number of i.i.d. copies, $n$ (Figure 1 of uploaded PDF).
>
> We did not intend to suggest that polarization occurs as $n$ increases if the channel varies with $n$, say by holding $nI(X;Y)$ fixed. In our experience, noisier channels polarize more slowly. If one increases $n$ and the channel noise simultaneously, it is unclear which of these effects will dominate.
>
> Thank you for highlighting various typos and errors. We will correct them in the final draft.

---

> > ### Comment · Reviewer_w9ei · 2024-08-10
> >
> > I thank the authors for their rebuttal. They have addressed my concerns, and I have updated my score accordingly.

---

### Official Review · Reviewer_Y8UR · 2024-06-20

**Soundness:** 2
**Presentation:** 1
**Contribution:** 2
**Rating:** 4
**Confidence:** 4

**Summary:**

The manuscript considers the design of algorithms for channel simulation. This topic has been extensively explored in the information theory literature, under various names including 'Reverse Shannon Theorems', 'Channel Synthesis', and 'Channel Simulation'.

The primary concern is that the problem formulation, proposed solution, and underlying applications do not align with the core focus areas of this conference, making it more appropriate for information theory and theoretical computer science venues. Additionally, the writing and presentation require significant improvements. The problem formulation is not rigorously provided, and many of the statements and proofs are incomplete and inaccurate. The proposed approach lacks significant novelty. Furthermore, the scope is restricted to the synthesis of symmetric binary-input channels.

**Strengths:**

The channel simulation problem is of significant interest in the information theory community both in the classical and quantum settings.

**Weaknesses:**

- The problem formulation, proposed solution, and underlying applications do not align with the core focus areas of this conference, making it more appropriate for information theory and theoretical computer science venues.
- The IID assumption on $X^n$ and $Y^n$, and the fact that only simulation of symmetric channels with binary-input is considered in the main body of the manuscript significantly limits the scope and applicability of the proposed algorithms.
- It is not clear why the main focus of the paper is on polar codes. As mentioned in the appendices, the ideas presented in the paper (which have roots in source coding literature) are applicable to linear codes in general. The authors provide examples of Trellis codes and general linear codes and lattices in the appendices, whereas a significant portion of the paper is focused on discussing the properties of polar codes.
- Common randomness is used to generate uniform variables $Z_i$ which are then used in the simulation protocol. It is not explained how the uniformly distributed $Z_i$ are produced. For instance, if the randomness is communicated through a discrete noiseless channel as in [1], or is available as a binary string [2], then the users cannot produce completely uniform $Z_i$ with limited computational complexity, which makes exact channel simulation using the proposed protocol impossible. This is not an issue in evaluating the fundamental limits of channel simulation as in [1,2] since computational complexity is not the focus of those works, and $Z_i$ can be made as close to uniform as necessary. However, in this work, which considers efficiency and algorithm design, the computational complexity of producing $Z_i$ must be discussed, and the model for common randomness available to the users be described in more detail. That is, given a fixed n, how close does $Z_i$ need to be to a uniform variable, and what is the complexity of producing such $Z_i$ from a common random string?
- Section III is not well-organized. The ideas are presented loosely and without formal introduction of the underlying concepts. The simulation of the reverse channel P_X|Y is described in detail, and it is stated that the scheme is used to simulate the actual channel $P_{Y|X}$ without further explanation. It is not clear why the description of the simulation protocol for $P_{Y|X}$ is not provided directly instead.

Minor Comments:

P.3 Line 99 -> blocklength m+n

Equations (21) and (22) -> $Z_i$ should be $Z_1$ and $Z_2$, respectively.


[1] Cuff, Paul. "Communication requirements for generating correlated random variables." 2008 IEEE International Symposium on Information Theory. IEEE, 2008.

[2]  Li, Cheuk Ting, and Abbas El Gamal. "Strong functional representation lemma and applications to coding theorems." IEEE Transactions on Information Theory 64.11 (2018): 6967-6978.

**Questions:**

- In Section III, the simulation of the reverse channel $P_{X|Y}$ is described in detail, and it is stated that the scheme is used to simulate the actual channel $P_{Y|X}$ without further explanation. Please explain the simulation procedure for the direct channel.
- Please explain why the paper focuses on polar codes as opposed to other linear codes, given the comparatively high decoding complexity of polar codes. The alternative approach, which is applicable to other linear codes is only discussed in the appendix. How do the approaches compare and why is preference given to the one utilizing polar codes in this work?

**Limitations:**

As mentioned in the manuscript, a major limitation of the approach utilizing polar codes is the restriction to binary-input symmetric channels.

---

> ### Author Rebuttal · Authors · 2024-08-07
>
> Thank you for your feedback. We address your comments and questions below:
>
> > ... This topic has been extensively explored in the information theory literature ...
>
> We disagree with this description. The information theory literature on this problem has predominantly focused on fundamental limits (i.e. theoretically achievable rates) without regard to computational complexity. Practically implementable schemes have mostly emerged from ML literature. See, e.g. [2-8].
>
> > ... the problem formulation, proposed solution, and underlying applications do not align with the core focus areas of this conference ...
>
> Again we disagree:
> * The *problem formulation* is drawn directly from the above papers
> * The *proposed solution* is more of interest to the ML community because it proposes a fast, implementable scheme
> * The *applications* mentioned in the second paragraph of the introduction are all drawn from machine learning, with citations to papers in ML venues.
>
> We are not aware of any work on channel simulation in the theoretical computer science community outside of [9], which is a decade old and focuses on fundamental limits, not practical schemes.
> > The problem formulation is not rigorously provided
>
> We feel the problem formulation in Section 2.1 is comparably rigorous to various papers in the ML literature on this exact problem, such as [3,4,7]
>
> It also matches that given in a more explicit form in many papers, e.g., [Section III, 10]; we can include such a formulation in the camera-ready version.
> > ... many of the statements and proofs are incomplete and inaccurate.
>
> There is only one short proof in the paper (Theorem 1 in Appendix C), which in our estimation, is entirely rigorous and correct. We would appreciate it if the reviewer could substantiate this comment by noting specific assertions that are incomplete or inaccurate.
> > The proposed approach lacks significant novelty.
>
> We view the use of error-correcting codes to achieve exact channel simulation as new.  Prior attempts have taken the form of hand-crafted solutions for small $n$ or variations on information-theoretic schemes --- completely different from PolarSim. We would appreciate it if the reviewer could substantiate this comment by indicating prior works with significant overlap.
> > ... restricted to the synthesis of symmetric binary-input channels
>
> The channel must be binary output, not binary input. Otherwise, we agree that this is a limitation of the paper. But, we note that
> 1. No existing scheme can achieve subexponential complexity for any nontrivial class of channels.
> 2. Other coding schemes, including variations of polar codes, can handle nonbinary channels. In particular, see Appendix A for a scheme that simulates a continuous channel.
> 3. For compression applications, binary-output channels are not unreasonable. Indeed, various papers have considered VAE-type architectures with discrete latents, including [11-13].
>
> In channel coding, no one code works optimally for all channels; different channels require different designs. It is natural to expect this to hold for channel simulation as well. Also, some existing schemes can only simulate a restricted class of channels [6].
>
> > The IID assumption ...
>
> As noted in the introduction, several applications of interest require the simulation of an i.i.d. channel with sizable $n$. Arguably one of the limitations of prior schemes is that they seek to simulate i.i.d. channels but fail to capitalize on the structure that this assumption entails. Far from being a weakness, we view one of the contributions of the paper as pointing out that an i.i.d. channel assumption is both reasonable and valuable.
>
> > Please explain why the paper focuses on polar codes ...
>
> The focus on polar codes is made clear in the introduction of the paper:
>
> "First, they [polar codes] have excellent channel coding performance,
> both theoretically (Mondelli et al. [2016]) and in practice (Egilmez et al. [2019]). Second, their complexity scales as $n \log n$. Third, they require no manual tuning. Fourth, they are simple to describe, requiring minimal background in coding theory."
>
> PolarSim relies on certain unique properties of polar codes; it does not extend to arbitrary linear codes.
>
> The schemes presented in the appendix have limitations: The dither-based approach in Appendix B can only simulate the BSC (because the input and output alphabets must be the same), and the trellis-based approximate AWGN simulator has no theoretical guarantees.
>
> Also, note that the decoding complexity of polar codes is $n \log n$. It is low, not high.
>
> > Common randomness ...
>
> The availability of infinite common randomness is the prevailing assumption in the ML literature on this problem. Determining the degree of "exactness" needed and the adequacy of pseudo-randomness are important and difficult problems, but they transcend this work and merit a separate paper.
>
> > Section III ...
>
> The beginning of Section 3 is focused on providing intuition for the main algorithm and hence written in an expository style while still preserving mathematical correctness. For a formal description, one should rely on Algorithms 1 and 2, Theorem 1, and the experimental results.
>
> > The simulation of the reverse channel ...
>
> The inherent duality between channel coding and channel simulation means that the way to use channel codes for simulation is to swap the roles of the encoder and the decoder, i.e., the channel coding decoder becomes the channel simulation encoder and vice versa. Thus, using the channel code for a channel $p_{Y|X}$, we can simulate the reverse channel $p_{X|Y}$.
>
> Given its fundamental nature, we chose not to hide this swap in our exposition. However, given that it has caused some confusion, in the final version we will develop the PolarSim scheme without highlighting this. Thank you for bringing this to our attention.
>
> We also thank you for pointing out typos and errors; we will correct them in the final draft.

---

> > ### Comment · Reviewer_Y8UR · 2024-08-08
> >
> > I thank the authors for their comprehensive response and their effort in addressing the comments from the previous round.
> >
> > Unfortunately, the response does not alleviate the concerns raised in the previous round. In particular, the following are my main remaining concerns:
> >
> > - The authors cite references [2-8] as examples of related prior works published in similar venues. While these works indeed focus on quantization, compression, and sampling—topics directly relevant to the NeurIPS audience—and use statistical techniques such as concentration of measure which are of great interest in this literature, the relevance of this paper to machine learning applications remains unclear. Although the authors argue that the channel simulation problem explored here can be viewed as a generalization of quantization, the applications of this generalization for ML scenarios are unclear. The paper's core methodologies, including polar coding and trellis codes, are more familiar to information theory and communication theory communities. Consequently, experts from these fields would be better positioned to evaluate the work's quality and novelty. The results would also be appreciated more if published in venues which focus on related research problems.
> >
> > - Many of the statements, including the problem formulation, are not presented precisely. The authors assert that Section 2.1 presents a rigorous problem formulation, it actually provides only a high-level introduction to the problem and existing literature. Definitions of concepts such as code, rate, and other relevant parameters should be given rigorously (see [10] Section III as an example). There are many other imprecise statements. For example, Equation (31) uses terms like "$\approx$" and "for most i" without clear definitions or quantifiable bounds. In a theoretical paper, such ambiguities significantly impact the work's rigor and overall quality, and make it difficult to verify the assertions. There are also typographical mistakes and undefined variables, which makes the paper less readable, for instance in Section 2.1,  'm and n can be combined to obtain a scheme for block length m + ny', the y should be removed. In Algorithm 2, `N' is not defined, etc.
> >
> > - The novelty of the work is not clear. The use of randomly generated codebooks for channel simulation is not novel (although using polar codes has not been specifically considered). For instance, please refer to the following paragraph in the literature review provided in [10]: `[prior] schemes follow the same general architecture—the common randomness is used to generate a large i.i.d. codebook containing different reconstruction strings, and the encoder stochastically selects a codeword and indexes it to the decoder.' Of course, the focus of those prior works was on deriving the fundamental performance limits, rather than constructive algorithms, thus they have focused on large i.i.d codebooks. The use of well-studied codes such as polar codes, in place of the large i.i.d. codebooks considered in those works, is an interesting directions, with limited novelty.
> >
> > - My concerns regarding the organization, especially the presentation of Section III have not been fully addressed. The authors mention that ` Section 3 is focused on providing intuition for the main algorithm and hence written in an expository style ' and that  modifications will be made in the final version of the work to improve some aspects of the presentation. However, given the significant changes required, I believe that the changes need to undergo a further round of review, once they are made, which is not possible given the current review process.

---

> > > ### Author Response · Authors · 2024-08-10
> > > **Thank you for your response. We respond to the concerns raised in your comment below.**
> > >
> > > > ... related prior works ...
> > >
> > > Our point in the rebuttal was not that works [2-8] are examples of tangentially related papers. Our point was that those papers (especially [3-8]) consider exactly the problem considered in this work: practically implementable schemes for channel simulation. In our view, this problem originated in the ML community, and most of the subsequent work on it has appeared in ML venues.
> > >
> > > > ... the applications ...
> > >
> > > The second paragraph of the introduction provides applications in machine learning including model compression [Havasi et al., 2018], federated learning [Shah et al., 2022], image compression with realism constraints [Theis et al., 2022], and VAE-based compression [Balle et al. 2020]. All of these papers apply channel simulation to an ML task, and would benefit from improved channel simulation methods. We could have provided many more papers in this list, but we felt that the channel simulation problem is now so firmly established in the ML community (see [3-8] above and the references therein) that this was unnecessary.
> > >
> > > > ... methodologies ...
> > >
> > > The paper's methodologies would indeed be more familiar to someone in information theory or communications. But the problem it is solving and the applications it impacts are more familiar to the ML community. We therefore believe that an ML venue is more appropriate for this paper. Indeed, many ML papers apply methodologies from statistics, information theory, or other disciplines. We view this as one of the strengths of ML as a field.
> > >
> > > One specific argument in favor of choosing an ML venue is that the prior state-of-the-art work [7] was published at last year's NeurIPS. The earlier state-of-the-art methods were also published in ML venues. It is not clear that reviewers of information theory or communications conferences would be aware of [7]. In contrast, some of the present reviewers are clearly familiar with that work.
> > >
> > > We also claim no contributions to the understanding of polar codes, which might be expected of an information theory paper. The coding theory background required to understand and evaluate our work is minimal, and we provide a self-contained description in our paper.
> > >
> > > > Many of the statements, ...
> > >
> > > We believe the formulation in Section 2.1 is perfectly rigorous. It is simply less explicit than, e.g., [10, Section III]. We provided the formulation in condensed form because we noticed that the problem had appeared in so many different ML papers over the years that some recent ML papers had described the problem in a similarly condensed form (as in, e.g., [7]).
> > >
> > > In any event, we believe there is no dispute about what the problem formulation is. The question is simply whether to state the problem with the level of detail of [7] (which appeared in NeurIPS last year) or [10] (which appeared in an information theory venue). Given the venue, we chose the former but it is trivial to replace this with the latter.
> > >
> > > > ... `N' is not defined, etc.
> > >
> > > The beginning of Section 3, as noted earlier, is meant to be expository and provide intuition. We have presented a formal statement and proof of correctness and optimality later in Section 3 and in Appendix C, which quantify all of the approximations mentioned. To verify the correctness of the assertions, one should rely on the theorem and the proof, not the informal discussion. We felt that some readers would find the paper more accessible if an intuitive discussion was provided in addition to the formal proof, especially since the typical NeurIPS reader might be unfamiliar with polar codes. Readers who are uncomfortable with this informality can proceed directly to the theorem and proof.
> > >
> > > 'N' is supposed to be 'n'; this is a typographical error. We thank the reviewer for pointing out this and the two typographical errors mentioned in the original review.
> > >
> > > > The novelty  ...
> > >
> > > We agree that most past works have used large i.i.d. codebooks. This is why they have exponential complexity. We believe that our polar-code-based scheme entails significant novelty for the following reasons:
> > > * Existing schemes have a complexity that scales exponentially with n. Ours has n log n complexity.
> > > * Our scheme is fundamentally different from past works. It has no notion of acceptance or rejection of samples. There is no codebook per se. Unlike past works, it does not aim for good performance for small n; Instead, it aims for scalability in n.
> > > * Our scheme significantly outperforms existing schemes in terms of rate.
> > >
> > > > ... the organization ...
> > >
> > > We said in the rebuttal that the beginning of Section 3 is written in an expository style. This serves as a warmup for the formal description that follows later in the section. We do not believe this section requires significant changes. We offered to edit the informal exposition to circumvent the channel “flip.” This essentially amounts to swapping the role of X and Y in the discussion, and we do not believe it merits additional rounds of review.

---

> > > > ### Comment · Reviewer_Y8UR · 2024-08-10
> > > >
> > > > I thank the authors for their comprehensive response and their effort in addressing the remaining concerns.
> > > >
> > > > The authors have explained that since the work focuses on practical implementation of channel simulation algorithms, rather than characterization of fundamental performance limits, it will be appreciated more in the ML community, which deals with the application of such techniques, rather than the IT community. This justification along with the discussion about the relevance to previously published work at Neurips addresses my concerns about relevance. I still have concerns about the presentation and rigor as outlined in previous comments.
> > > >
> > > > I have updated my score accordingly to reflect the fact that some of my major concerns have been addressed.

---

> > > > > ### Author Response · Authors · 2024-08-11
> > > > > **On Presentatoin and Rigor**
> > > > >
> > > > > We'd like to thank the reviewer for their engagement with this process. We appreciate it.
> > > > >
> > > > > Would the reviewer's concerns about presentation and rigor be addressed if at the start of Section 3 there was a statement explaining that the first part of the section provides an intuitive exposition of the idea, with the formal statement of the algorithm and guarantees coming afterward? We feel that providing a non-rigorous development of the ideas improves readability, especially since the typical NeurIPS reader might not be familiar with error correcting codes. It is easier to grasp the high-level idea if the discussion can sidestep the technical details. At the same time, we feel that including such an intuitive exposition does not reduce the rigor of the paper so long as the assertions are backed up with formal statements and proofs later, as is the case here.
> > > > >
> > > > > Including such a statement would make clear to readers that a rigorous development is available, and it would help them find it. This would represent a minor edit.

---

> > > > > > ### Comment · Reviewer_Y8UR · 2024-08-13
> > > > > >
> > > > > > I thank the authors for their response and effort in improving the manuscript.
> > > > > >
> > > > > > Regarding the latest response about concerns with presentation and rigor, unfortunately, the presentation of the paper requires significant improvements in my view. For instance, the other reviewers and I have pointed out a number of typographical issues. Similarly, the content needs to be more rigorous and better organized as discussed in my previous comments. I greatly appreciate the authors' willingness to improve the presentation of the manuscript, and understand their justification that in their view the high level discussions are helpful for some readers, although I disagree with the choice of style personally. At the same time, I believe that the required changes and corrections would be significant enough to merit another round of review, which is not possible due to the current review process.
> > > > > >
> > > > > > Consequently, I will keep my current score. However, since my other concerns have been addressed, if the other reviewers have a positive view of the manuscript and overall recommend its acceptance, I will not be opposed to the acceptance of the manuscript.

---

### Official Review · Reviewer_6bFv · 2024-07-11

**Soundness:** 3
**Presentation:** 2
**Contribution:** 3
**Rating:** 4
**Confidence:** 2

**Summary:**

This paper considers the scalability problem in the channel simulation. Channel coding, specifically polar coding, is introduced to improve the performance of channel simulation. The topic is interesting and the work is valuable.

**Strengths:**

This paper uses error correction codes to improve the channel simulation with significant results.

**Weaknesses:**

The paper stresses "fast" in the title, but the corresponding justification or even a statement is missing in the main body of the paper. Some notations are not well defined. For example, what is the meaning of performance? It seems to be the rate, but it also seems to be speed (according to the title of this paper).

A table is needed to compare the results with the other works.

**Questions:**

What is the meaning of the performance?

How to justify "fast" in the title?

**Limitations:**

Yes.

---

> ### Author Rebuttal · Authors · 2024-08-07
>
> Thank you for your feedback and comments. We have addressed specific comments and questions below.
>
> > The paper stresses ``fast'' in the title, but the corresponding justification or even a statement is missing in the main body of the paper. Some notations are not well defined. For example, what is the meaning of performance? It seems to be the rate, but it also seems to be speed (according to the title of this paper).
>
> > What is the meaning of the performance?
>
> > How to justify "fast'' in the title?
>
> We are interested in both (data) rate and (execution) speed, and, per (1) and (3) in the introduction, the two are coupled: a faster algorithm allows one to handle larger $n$, which improves the rate. "Performance'' refers to the rate $R_n$ defined in the introduction, i.e., the average number of bits transmitted divided by the number of times the channel is being simulated.
>
> We felt that the term "fast'' in the title was self-explanatory given that $\mathtt{PolarSim}$ has $n \log n$ complexity while (quoting from the paper) "there are currently no known schemes that simulate any nontrivial class of channels with even subexponential complexity in $n$."
>
> The speed of the scheme is what enables the improved rates that we observe in Figure. 3 (See Figure 2 of attached PDF for updated version)

---

> > ### Comment · Reviewer_6bFv · 2024-08-09
> > **I thank the authors for their comprehensive response.**
> >
> > The contributions should be clearly and explicitly stated and compared in the main body of paper, rather than self-explanatory and deduced by the readers. The efficiency should be compared with the other works in terms of theoretic analysis and experiments in details.

---

> > > ### Author Response · Authors · 2024-08-13
> > > **Runtime Comparison**
> > >
> > > Thank you for your comment.
> > >
> > > The paper does include a theoretical comparison of the complexity: existing schemes such as PFRL and GPRS have complexity that is exponential in $n$, whereas PolarSim's is n log n. We did not compare the runtimes experimentally because we felt the vast gulf in their theoretical complexities rendered this unnecessary. But it is simple to perform such a comparison. For the setup in Fig. 3, these are the run-times in seconds, averaged over 50 runs:
> > >
> > > | p     | GPRS   | PolarSim |
> > > |-------|--------|----------|
> > > | 0.441 | 40.570 | 0.098    |
> > > | 0.299 | 57.623 | 0.120    |
> > > | 0.225 | 61.514 | 0.110    |
> > > | 0.171 | 69.175 | 0.110    |
> > > | 0.127 | 75.816 | 0.114    |
> > > | 0.091 | 67.527 | 0.122    |
> > > | 0.061 | 72.374 | 0.123    |
> > > | 0.035 | 71.960 | 0.118    |
> > > | 0.015 | 64.417 | 0.116    |
> > > | 0.000 | 59.206 | 0.115    |
> > >
> > > Here $p$ is the crossover probability of the channel. PolarSim is operating directly on the $n = 1024$ copies of the channel (and takes about $.1$ seconds per run to do so). For GPRS, we run the algorithm separately on $128$ blocks of size $8$ (so that effectively $n = 8$). This takes about $70$ seconds. Thus, PolarSim is over $600$x faster than GPRS while its data rate is significantly better, as shown in Fig. 3.
> > >
> > > The GPRS implementation has been optimized according to the suggestion from reviewer w9ei. For this particular setup, this GPRS implementation could be parallelized by running the $128$ blocks (or a subset thereof) simultaneously. Thus, the runtime of GPRS could be reduced. On the other hand, the complexity of GPRS is still exponential in $n$, so running on $114$ blocks of size $9$ would result in a substantial slow down.
> > >
> > > Such a table could be easily included in Fig. 3., alongside the existing plot. We hope this addresses the concern about the efficiency comparison.

---

### Official Review · Reviewer_y5AJ · 2024-07-11

**Soundness:** 4
**Presentation:** 4
**Contribution:** 3
**Rating:** 8
**Confidence:** 1

**Summary:**

The paper studies the channel simulation problem, whose goal is to minimize the number of transmitted bits so that the decoder can generate an output according to a target distribution given the encoder’s input. The paper proposes a channel simulation method, called PolarSim, based on polar codes. The rate of PolarSim approaches the mutual information lower bound by scaling up the block length, as theoretically proven and demonstrated by experiments.

**Strengths:**

1.	The authors propose a novel method for channel simulation.
2.	The paper is technically sound (to the best of my knowledge) and well-written.
3.	The performance of the proposed method is impressive: The experiment results demonstrate that the proposed method achieves a rate approaching the theoretical bound, outperforming a quiet recent method.

**Weaknesses:**

1.	The smallest $n$ used in the experiments in the paper is $1024$. How would PolarSim perform for smaller $n$? Could the authors provide some figures in the supplementary material or at least comment on this?
2.	In Figure 3, the authors mention that “Data for GRPS is only plotted for parameters where the algorithm consistently terminates in a reasonable amount of time.” It is unclear what the authors mean by “a reasonable amount of time.” Can the authors be more explicit about what they mean here?

Minor comments and typos:

1.	The figures would be easier to read if the colors were included in a legend within the plot.
2.	Line 55: Seems like “means that” should be “this means that.”
3.	In line 118, I would suggest the authors to define the $F_2$ notation.
4.	Line 124: “likeihood” should be “likelihood.”
5.	Line 125: “thatl” should be “that.”
6.	Line 227: “)” is missing.

**Questions:**

- Could the authors repeat their contributions in Conclusions so the contributions are clearer?
- Please see Weaknesses for a few more questions.

**Limitations:**

The authors have explicitly addressed the limitations of their method.

---

> ### Author Rebuttal · Authors · 2024-08-07
>
> Thank you for your encouraging assessment of our work and for the comments and feedback you have provided. We have addressed specific comments and questions below.
>
> > The smallest used in the experiments in the paper is 1024. How would $\mathtt{PolarSim}$ perform for smaller $n$? Could the authors provide some figures in the supplementary material or at least comment on this?
>
> In our rebuttal PDF, we have provided a plot of the redundancy (meaning the rate minus the mutual information) versus $n$ for the $\mathsf{BSC}, \mathsf{BEC}$ and the $\mathsf{AWGN}$ (Please refer to Figure 1. of the attached PDF). There we see that the $\mathtt{PolarSim}$ scheme performs well even for small values of $n$. For comparison, we also plot the achievable rate from [Eq (1),1] (i.e. the upper bound), labeled in the plots as "PFR UB". We will include this in the camera-ready version of our paper, likely in the appendix.
>
> > In Figure 3, the authors mention that “Data for GRPS is only plotted for parameters where the algorithm consistently terminates in a reasonable amount of time.” It is unclear what the authors mean by “a reasonable amount of time.” Can the authors be more explicit about what they mean here?
>
> We were facing issues with getting the GPRS algorithm to terminate in high mutual information regimes during our experiments. However, we have followed the suggestions of reviewer w9ei to speed up the implementation of the GPRS algorithm and as a result, this issue is no longer a concern. We have presented the updated plot for Figure 3 (Please refer to Figure 2 in the attached PDF) which we will include in the camera-ready version of the paper.
>
> > Could the authors repeat their contributions in Conclusions so the contributions are clearer?
>
> Thank you for this suggestion. We will take this into consideration when preparing the camera-ready version.
>
> > The figures would be easier to read if the colors were included in a legend within the plot.
>
> Our concern here was potentially overcrowding our plots. That said, we will try our best to make the suggested change work and include it in the camera-ready version.
>
> We also thank you for pointing out certain errors and typos in our manuscript. We will be sure to incorporate these changes in our camera-ready version.

---

> > ### Comment · Reviewer_y5AJ · 2024-08-08
> >
> > Thank you for addressing my comments.

---

### Author Rebuttal · Authors · 2024-08-07

We thank all the reviewers for taking the time to review our paper. Their feedback and suggestions will be valuable in helping us refine our work.

Based on their comments, these emerged as the main areas for improvement for our paper:
* Studying the performance of $\mathtt{PolarSim}$ for small $n$ .
* Simplifying our description in Section 3. to avoid any potential confusion regarding the directionality of the simulated channel
* Our comparison plot (Figure 3 in the paper)
   * Providing full details of our experiments for greater reproducibility
   * investigating/fixing the convergence issues for GPRS
* Providing full experimental details for the scheme presented in Appendix A

 We have detailed our strategy to tackle these issues in our responses to the individual reviewers. We have also addressed other questions and clarifications that were raised in the reviews.

# Supplementary PDF
We have also included a one-page PDF attachment that contains the following figures
* Figure 1 contains plots of the normalized redundancy $R_n - I(X;Y)$ vs the blocklength $n$ for three different channels.
* Figure 2 contains an updated plot of Figure 3 from the paper, based on the suggestions offered by Reviewer w9ei.

# References
1. Cheuk Ting Li and Abbas El Gamal, "Strong functional representation lemma and applications to coding theorems", *IEEE Transactions on Information Theory*, 2018.
2. Chris J. Maddison, Daniel Tarlow, and Tom Minka, "$A^*$ sampling,'' *NeurIPS*, 2014.
3. Eirikur Agustsson and Lucas Theis, "Universally quantized neural compression,'' *NeurIPS*, 2020.
4. Gergely Flamich, Stratis Markou, and José Miguel Hernández-Lobato, "Fast relative entropy coding with $A^*$ coding,'' *ICML*, 2022.
5. Lucas Theis and Noureldin Yosri, "Algorithms for the communication of samples,'' *ICML*, 2022.
6. Gergely Flamich, Stratis Markou, and José Miguel Hernández-Lobato, "Faster relative entropy coding with greedy rejection coding.'' *NeurIPS*, 2023.
7. Gergely Flamich, "Greedy Poisson rejection sampling,'' *NeurIPS*, 2024.
8. Buu Phan, Ashish Khisti, and Christos Louizos, "Importance matching lemma for lossy compression with side information,'' *AISTATS*, 2024.
9. Braverman and Garg, "Public vs private coin in bounded-round information'', *ICALP*, 2014
10. Sharang M Sriramu and Aaron B Wagner, "Optimal redundancy in exact channel synthesis", *arXiv
preprint*, 2024.
11. J. T. Rolfe, "Discrete Variational Autoencoders'', *ICLR*,  2017
12. J. Fajtl, V. Argyriou, D. Monekosso and P. Remagnino, "Latent Bernoulli Autoencoder'', *ICML*, 2020.
13. E. Özyilkan, J. Ballé, and E. Erkip, “Learned Wyner–Ziv compressors recover binning”, *ISIT*, 2023.
14. David S Taubman, Michael W Marcellin, "JPEG2000: Image compression fundamentals, standards and practice", *Springer*, 2002.

---

### Decision · Program_Chairs · 2024-09-25

**Decision:**

Accept (poster)

**Comment:**

This paper proposes a practically-implementable scheme for channel simulation, which uses polar codes.

The rating/confidence scores of the reviewers were 8/1, 4/2, 4/4, and 7/4, showing a relatively large split.
I would like to note that such a large split is not uncommon for papers on interdisciplinary topics, like this one.
I have read this paper myself, as well as the reviews and discussion with the authors. Besides the clarity issues and the weakness in the experimental section (they were on a few basic channel models, and no machine-learning related experiments were conducted), as raised by some reviewers, I have a few additional concerns:
- **Polar codes:** The proposed scheme does not use polar codes but just channel polarization, or more precisely, channel polarization in terms of mutual information.
I think that this point is important, because it makes the successive-cancellation (SC) decoding and the rate of polarization unnecessary in this paper, although they are needed to prove the capacity-achieving property (i.e., that the block-error rate can be made arbitrarily small as the codelength tends to infinity) of polar codes. Clarification on this point will avoid possible misunderstanding (e.g., why do not existing non-polarizing channels when $n$ is not large enough cause any problem, why is the stochastic sampling (SoftPolarDec) used instead of SC decoding, etc.) of those who are knowledgeable about polar codes.
- **Experiments:** My understanding of the proposed scheme is: Given a channel, one first performs channel polarization transform recursively to obtain $n$ polarized subchannels. One then evaluates the subchannel parameters $\bar{p}^n$. The rate is then determined *a posteriori* via e.g. arithmetic coding. In Figure 2, however, the parameters to be adjusted are shown in the vertical axes, and the resulting rates are shown in the horizontal axes, which is just the opposite of the common convention of taking the independent variable in the horizontal axis and the dependent variable in the vertical axis. I also guess that the percentiles shown in Figure 2 result from the variable-length nature of arithmetic coding, which is not explained explicitly.

After the discussion among us did not resolve the large split in the scores, but the agreement we have reached is that the significance would outweigh those weaknesses raised by the reviewers. I thus would like to recommend acceptance of this paper, on the premise that the points raised by the reviewers will be properly addressed in the final version.

Minor points:
- Line 21: or (or)
- Line 55: (it) means that
- Line 79: What does "balance" mean?
- Line 85: The notation $p^{\times n}$ using the superscript "$\times n$" is not restricted to the channels, as can be seen on line 93, where it is applied to the input distribution $p_X$.
- Line 89: 1 → $p$
- Line 99: $m+n\mathrm{y}$ → $m+n$
- Line 123: "accordingly" is not clear enough; that(l)
- Line 128: $0$, $1/2$, or $1$ → $0$, $1$, or $\infty$
- Lines 142-147: Under the channel model assumed here, $(X_i,Y_i)$ are not iid, so that, strictly speaking, one cannot apply equation (3) directly here.
- Line 152: It would be better if the authors introduce the symbols $V_i$ and $Δ_i$ to denote a common bit and a difference bit, respectively.
- Line 154: suffices to send (it) $Δ^n$
- Equations (21) and (22): $Z_i$ → $Z_1$ or $Z_2$
- Equation (22): Whether the value of $U_1$ generated according to equation (21) is to be used as $u_1$ appearing in the conditioning part of equation (22) is not clear.
- Line 187: in (in)
- Algorithm 2: $N$ → $n$
- Line 227: $N(0,\sigma^2$ → $N(0,\sigma^2)$
- Line 231: mutual information lower (bound)
- Line 398: a AWGNchannel → an AWGN channel
- Line 400: developed (by) as
- Line 407: $\{1\ldots k\}$ → $\{1,\ldots,k\}$
- Equation (54): Some explanation on how to derive it using equations (48)-(52) would be beneficial, as it may be confused with "mismatched entropy", represented as a sum of the true entropy and the KL divergence.